# VISUALTHINKER: R1-ZERO'S "AHA MOMENT" IN VISUAL REASONING ON A 2B NON-SFT MODEL

## ABSTRACT

The recent DeepSeek-R1 demonstrated how reinforcement learning with simple rule-based reward can enable autonomous development of complex reasoning in large language models, characterized by the "aha moment", in which the model manifest self-reflection and increased response length during training. However, attempts to extend this success to multimodal reasoning often failed to reproduce these key characteristics. In this study, we present the first successful replication of these emergent characteristics for multimodal reasoning on only a non-SFT 2B model. Starting with Qwen2-VL-2B and applying reinforcement learning directly on the SAT dataset, our model achieves **59.47%** accuracy on CVBench, outperforming the base model by approximately **~30%** and exceeding SFT setting by **~2%**. By further incorporating a small amount of cold-start data, we achieved **70.58%** accuracy on CVBench, a performance surpass GPT-4o-mini. In addition, we observed that applying RL to instruct models often leads to trivial and low-diversity reasoning trajectories and presents our insights and attempts to understand and mitigate this issue.

## 1 INTRODUCTION

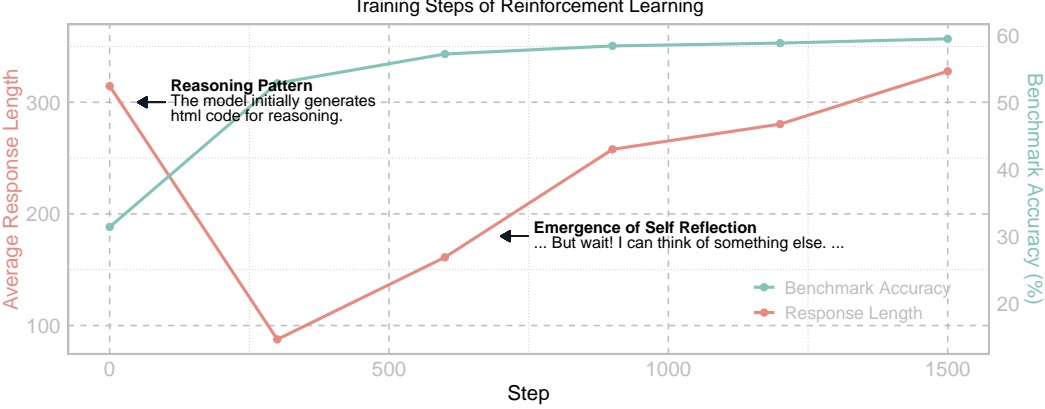

Figure 1: **The training dynamics of VisualThinker-R1-Zero on Qwen2-VL-2B base model**. Benchmark accuracy is measured on CV-Bench, and the average response length is calculated from rolling outs on SAT training samples. Initially, we observed a drop in length because the base model tended to generate HTML code. This behavior was quickly suppressed by RL, leading the model to adopt a more appropriate output format and a regular increase in response length. Afterwards, we observed a multimodal 'aha moment'—the emergence of self-reflection in models' response, as described in the DeepSeek-R1 paper, followed by a consistent positive correlation between response length and benchmark accuracy.

Recently, DeepSeek R1 [5] has demonstrated how Reinforcement Learning (RL) with simple rule-based incentives can enable a Large Language Model (LLM) to build complex reasoning capabilities autonomously. A key finding from this work was the emergence of advanced reasoning patterns

without explicit supervision—what the researchers termed the "aha moment," characterized by self-reflection and spontaneous increase in response length during training as the model learned to explore increasingly sophisticated problem-solving strategies.

Many researchers [3; 37; 24; 30] have attempted to extend this success to multimodal reasoning. However, most of these efforts have primarily struggled to reproduce the key characteristics exhibited by DeepSeek R1 mentioned above—specifically, the emergent "aha moment" and increased response length during reasoning. These implementations also failed to demonstrate the autonomous development of sophisticated reasoning strategies observed in DeepSeek R1's training.

In this work, we present **the first successful replication** of these key characteristics for multimodal reasoning on **only a non-SFT 2B model**. With both "aha moment" and increased length (Figure 1), our approach demonstrates that direct application of reinforcement learning on non-sft model can sufficiently induce sophisticated reasoning capabilities even in smaller multimodal models without supervised fine-tuning. We start from the Qwen2-VL-2B [29] base model and directly perform reinforcement learning. Without any SFT, our model achieves **59.47% accuracy on CVBench** [27], outperforming the base model by approximately **~30%** and exceeding the SFT model by **~2%**. Inspired by [5], we further present **VisualThinker R1**, a derivative initialized with a small amount of reasoning trajectories curated from **VisualThinker R1-Zero** and existing visual reasoning trajectory dataset to ease RL cold-start instability while keeping the generation diversity typically lost in large-scale supervised fine-tuning, achieving **70.58% accuracy on CVBench**, comparable to GPT4o.

In addition, we share our insights and failed attempts in achieving R1-like visual reasoning using RL with instruct models. We observed that starting from a supervised fine-tuned model often failed to reproduce the observations and findings reported by DeepSeek-R1. Upon investigation on this issue, we found that (1) Despite improved performance, RL on instruct model leads to **trivial reasoning** rather than genuine problem-solving strategies, and (2) freezing visual components or naive length reward are ineffective at inducing deeper reasoning capabilities. Our study characterizes RL's behavior in visual-centric tasks and provide **the first clear evidence that emergent multimodal reasoning can arise under the right conditions**. We believe these findings open a promising direction for studying multimodal reasoning dynamics and lay the groundwork for future work seeking to induce and understand emergent reasoning in multimodal setting.

## 2 RELATED WORK

### 2.1 MULTIMODAL REASONING

Researchers have demonstrated that LLMs can be post-trained to elicit enhanced reasoning abilities [16; 31]. With pre-trained visual encoders to understand visual content alongside textual data, multimodal LLM's reasoning abilities are investigated and enhancement typically requires sophisticated prompting designs [11] or large amounts of reasoning training data [26; 33]. The research community is increasingly interested in developing more natural methods to incentivize higher intelligence in models without relying on extensive supervised data or complex prompting techniques.

### 2.2 THE "AHA MOMENTS" IN DEEPSEEK R1

A recent breakthrough study, DeepSeek R1 [5], demonstrated that reinforcement learning can incentivize a model's reasoning abilities without any supervised reasoning data. Intriguingly, researchers [5] discovered an "aha moment" when directly applying RL with rule-based reward on mathematical datasets—a qualitative behavioral shift during RL training in which the model begins to autonomously exhibit more sophisticated reasoning behaviors such as revisiting earlier steps, expressing uncertainty, and performing self-correction.

We summarize the key characteristic of DeepSeek R1 and compare them with our model and other multimodal replications in Table 1. Specifically, we highlight two emergent phenomena: the "aha moment" and increasing response length.

The "aha moment" refers to the model's autonomous development of advanced problem-solving strategies during training, while the increasing response length indicates the model naturally learns to allocate more thinking time for reasoning tasks.

It remains questionable whether existing multimodal replications [9; 3; 8] without these key characteristics can be considered truly valid implementations of the R1-Zero approach.

Following our first replication of multimodal "aha moment", several works attempt to elicit sophisticated reasoning patterns in VLMs through alternative approaches. LMM-R1 [19] introduced a two-stage training pipeline that transfers textual reasoning ability to multimodal reasoning ability. Vision-R1 [12] incorporated high-quality data as cold-start to overcome the challenges of trivial reasoning behavior. MM-EUREKA [15] further extends these explorations by scaling the cold-start dataset and model size. Other subsequent works explore similar directions towards visual reasoning by incorporating diverse data [34], explicitly enhancing perception [4], encouraging rethinking strategy [28] applying iteration of training [7].

Table 1: **Comparison between DeepSeek R1 and its multimodal replications**. VisualThinker-R1-Zero is the first model that replicates the two key chararistics of DeepSeek R1 on vision tasks - the emergent "aha moment" and increased response length during reasoning.

| Feature | DeepSeek R1-Zero [5] | VisualThinker R1 Zero (Ours) | R1-V [3] | R1-Multimodal-Journey [9] | open-r1-multimodal [8] |
|---|---|---|---|---|---|
| Base Model | DeepSeek V3 | Qwen2-VL-2B | Qwen2-VL-2B-Instruct | Qwen2-VL-2B-Instruct | Qwen2-VL-2B/7B-Instruct |
| Modality | Language | Vision + Language | Vision + Language | Vision + Language | Vision + Language |
| Aha Moment | Yes | Yes | No | Yes | No |
| Response Length Dynamics | $\uparrow$ | $\uparrow$ | $\downarrow$ | $\downarrow$ | $\downarrow$ |
| Supervised Fine-tuning? | No | No | Yes | Yes | Yes |

# 3 VISUALTHINKER

In this section, we first introduce the training recipe for VisualThinker-R1-Zero, where a small non-SFT base model exhibits the "aha moment" when we directly apply RL to the model. To mitigate cold-start instability in RL, we further present VisualThinker-R1, a model initialized with a small set of reasoning trajectories sampled from VisualThinker-R1-Zero and existing visual reasoning datasets.

**Base Model**   Our method builds on Qwen-2-VL-2B [29] as the base model, applying GRPO [23] with a tailored chat template and prompting strategy to enhance its reasoning capabilities. We argue that applying GRPO to the base model is a more efficient and effective approach to replicate multimodal R1-style reasoning, as the base model preserves greater generation diversity. As shown in Section 5.1, the instruct variant, Qwen-2-VL-2B-Instruct, requires substantially more training resources and still fails to reproduce the "aha moment", exhibiting several notable failure modes.

**Training Recipe**   We train our models on **SAT** [21], a VQA dataset comprising 218k question-answer pairs synthesized using a photo-realistic physics engine to enhance spatial intelligence. Our training focuses on the static subset, which includes questions on relative spatial relationships, relative depth, and object counting. To let the base model explore various spatial reasoning for each question $q$ in the dataset $Q$, we apply the following chat template:

> **Prompt Template**
>
> A conversation between User and Assistant. The user asks a question about the image, and the Assistant solves it. The assistant first thinks about the reasoning process in the mind and then provides the user with the answer.\n User: {QUESTION} \n Assistant: Let me solve this step by step.\n <think>

For each question $q$ in the dataset, the model generates a response $o$ using this prompt template, and it is then optimized using the RL objective.

**RL algorithm**   Existing attempts applying RL on top of fine-tuned visual models failed to replicate DeepSeek r1's key characteristics. In contrast, we witness **prolonged reasoning trajectory** and **"aha moment"** with an **overlooked approach that directly applies GRPO [23] to an non-SFT model.** Our findings suggest that this setting is the key to true "aha moment" in multimodal reasoning. Now we briefly review the GRPO algorithm we adopted for RL training.

To reduce the overhead of training an additional value function model as required by PPO [22], GRPO uses the average reward of sampled responses from the policy model as a baseline for

computing the advantage. Specifically, given an input question $q$, we first sample a group of responses $\{o_1, o_2, \cdots, o_G\}$ and compute their corresponding rewards $\{r_1, r_2, \cdots, r_G\}$ with the reward model. The advantage is then computed as:

$$\hat{A}_{i,t} = \widetilde{r}_i = \frac{r_i - \text{mean}(\mathbf{r})}{\text{std}(\mathbf{r})}.$$ (1)

The policy model is then optimized by maximizing the following KL objective:

$$\mathcal{J}_{GRPO}(\theta) = \mathbb{E}_{q \sim P(Q), \{o_i\}_{i=1}^G \sim \pi_{\theta_{old}}(O|q)} \left[ \frac{1}{G} \sum_{i=1}^G \frac{1}{|o_i|} \sum_{t=1}^{|o_i|} \left\{ \min \left[ \frac{\pi_\theta(o_{i,t}|q, o_{i,<t})}{\pi_{\theta_{old}}(o_{i,t}|q, o_{i,<t})} \hat{A}_{i,t}, \right. \right. \right.$$
$$\left. \left. \left. \text{clip} \left( \frac{\pi_\theta(o_{i,t}|q, o_{i,<t})}{\pi_{\theta_{old}}(o_{i,t}|q, o_{i,<t})}, 1 - \epsilon, 1 + \epsilon \right) \hat{A}_{i,t} \right] - \beta \mathbb{D}_{KL} \left[ \pi_\theta || \pi_{ref} \right] \right\} \right],$$ (2)

where $\pi_\theta$ and $\pi_{old}$ are the current and old policy, and $\epsilon$ and $\beta$ are hyper-parameters introduced in PPO.

**Reward Modeling**  Following DeepSeek-R1, our RL approach remains elegant, avoiding the use of reward models [18] or Monte Carlo tree search (MCTS)-like techniques [35]. Specifically, we employ a rule-based reward function that evaluates responses based on their format and correctness:

- If the response provides a final answer and is correct, the model receives an accuracy reward of +1.

- If the response encloses its thinking in `<think></think>` and the final answer in `<answer></answer>` tags, the model receives a format reward of +1.

- otherwise, the model receives 0 reward.

Initial experiments suggest that this reward function helps the policy model quickly converge towards generating responses in the desired format.

**VisualThinker R1**  Inspired by the design in DeepSeek R1 [5], we introduce **VisualThinker R1**, a variant initialized with a small number of curated reasoning trajectories to ease the cold start instability of RL training while preserving the generation diversity often lost in large-scale supervised fine-tuning. Specifically, we use 7K long-chain-of-thought samples drawn from VisualThinker R1-Zero's trajectories on the SAT dataset and the R1-Distilled Visual Reasoning Dataset adapted from R1-V [3]. We furthr include a limited set of examples challenging dynamic visual reasoning (e.g. egocentric movement, object movement) from SAT  [20] during the reinforcement learning stage. Together, these components form our final training recipe.

## 4 EXPERIMENTS

In this study, we demonstrate that small non-sft model could solve vision-centric spatial reasoning task with our method, a type of benchmark often pose challenges to even larger models.

### 4.1 EVALUATION

To evaluate the generalization of our method, we use three existing spatial bench-marks—CVBench [27], BLINK (spatial subsets) [10], and Visual Spatial Relations (VSR) [13]. These vision-centric benchmarks formulate natural language questions to assess models' abilities in spatial relationships, object counting, depth ordering, and relative distance. This setup allows us to systematically examine how well our recipe could improve spatial reasoning capabilities.

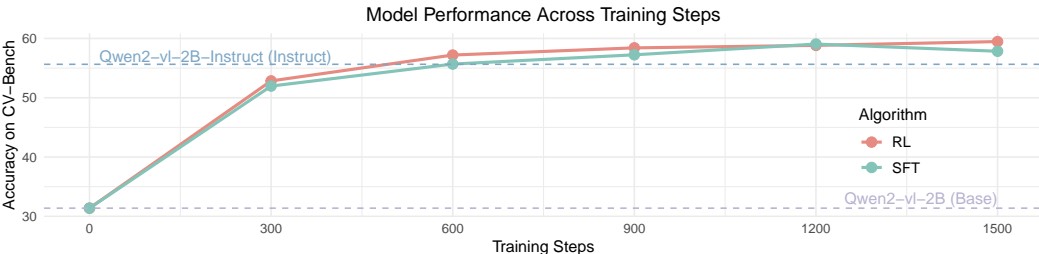

Figure 2: **Comparison between RL and SFT training.** Our method achieves a significant improvement over the base model and the instruction fine-tuned model. Specifically, Qwen2-VL-2B + R1 outperforms Qwen2-VL-2B (base model) by approximately ~30 and Qwen2-VL-2B-Instruct (instruction fine-tuned model) by ~5%, and Qwen2-VL-2B SFT (base model + SFT) by ~2%.

Table 2: **Accuracy (%) on vision-centric benchmarks**. The **bold** and underline values indicate the highest and second-highest scores among the open-source models.

| Model | Count | Relation | Depth | Distance | CV Avg | RelDepth | SpatRel | BLINK Avg | VSR Avg |
|---|---|---|---|---|---|---|---|---|---|
| *Proprietary Models* | | | | | | | | | |
| GPT-4o [17] | 70.43 | 80.46 | 81.50 | 80.00 | 77.59 | 72.58 | 85.31 | 79.40 | 80.27 |
| GPT-4o-mini [17] | 63.45 | 80.61 | 61.33 | 71.16 | 68.95 | 61.29 | 76.92 | 69.66 | 68.51 |
| *Open-Sourced Reasoning Models* | | | | | | | | | |
| VLAA-Thinker-Qwen2VL-2B [2] | 25.88 | 32.61 | 58.50 | 44.66 | 39.23 | 55.64 | **65.73** | 61.04 | 59.68 |
| VLM-R1 (OVD) [3] | 51.52 | 51.85 | 52.50 | 44.17 | 50.15 | 40.32 | 2.80 | 20.22 | **80.77** |
| *Baselines* | | | | | | | | | |
| Qwen2-VL-2B [29] | 54.69 | 22.46 | 0.16 | 31.66 | 31.38 | 13.70 | 0.69 | 6.74 | 0.00 |
| Qwen2-VL-2B + SFT | 60.02 | 68.92 | 55.00 | 45.83 | 57.84 | 58.06 | 47.55 | 52.43 | 35.80 |
| *Ours* | | | | | | | | | |
| VisualThinker R1-Zero | 59.64 | 66.76 | 54.16 | 56.66 | 59.47 | 50.80 | 55.94 | 53.18 | 62.32 |
| VisualThinker R1 | **63.32** | **79.08** | **75.00** | **66.50** | **70.58** | **61.29** | 63.63 | **62.54** | 63.69 |

## 4.2 Highlighted Results and Findings

**RL on base model autonomously developing longer responses correlated with improved performance on CV-Bench** We compare our method with the non-sft pre-trained model, Qwen2-VL-2B as the baseline on CV-Bench. During training, we observed that the model autonomously generated increasingly longer responses, accompanied by improved performance, as shown in Figure 1. Specifically, VisualThinker-R1-Zero outperforms Qwen2-VL-2B (base model) by approximately ~30 and Qwen2-VL-2B-Instruct (instruction fine-tuned model) by ~5%.

**RL demonstrates superior performance compared to SFT** We further compare our method against the base model SFT on the same SAT dataset. Our experimental results clearly demonstrate that applying reinforcement learning directly to the base model yields superior performance compared to traditional supervised fine-tuning methods, with an improvement of approximately 2% as illustrated in Figure 2.

**RL can help model improve general performance across vision-centric datasets** Apart from CVBench, we tested our method on various spatial reasoning datasets, including BLINK and VSR, illustrated in Table 2. Our method demonstrated improved performance over the Qwen2-VL-2B (base) by ~30%, and the Qwen2-VL-2B SFT (base + SFT) by ~2% on CV-Bench. On the BLINK and VSR benchmark, our method also achieves around 27% advantage comparing against the model trained with SFT. This suggests that visual reasoning could significantly benefit from R1-Zero training, demonstrating more scalable training through RL's exploration of diverse reasoning.

**Cold-start data incorporation further enhances performance** We discovered that VisualThinker R1-Zero's performance could be further improved by approximately 11% through the strategic incorporation of a small amount of cold start reasoning trajectories and dynamic visual capability challenging samples. This optimization resulted in a total accuracy of 70.58% (VisualThinker R1), a level comparable to state-of-the-art proprietary models such as GPT-4o-mini.

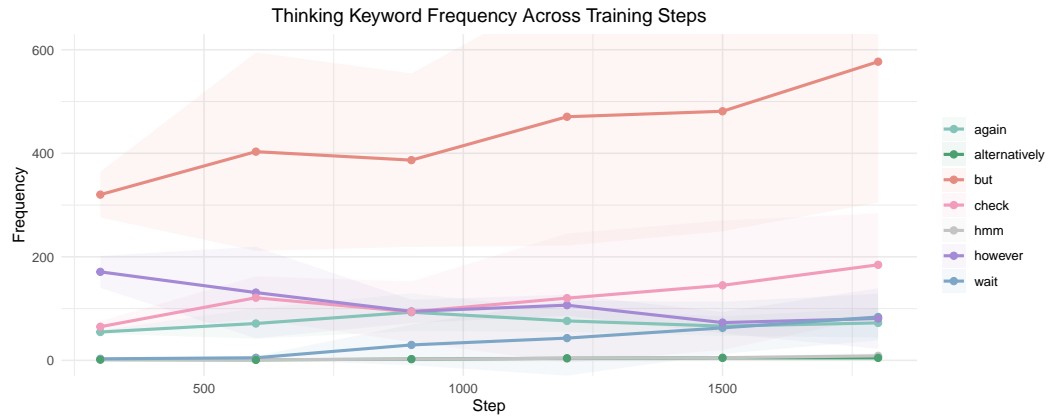

Figure 3: **Statistical analysis of reasoning-indicative keywords ("wait," "again", "but", "however", "so", "hmm", "check", "alternatively", and "mistake") across training steps in multi runs.** The results highlight the emergence of reflection and refinement behaviors after 600 steps, followed by stabilization of reasoning patterns beyond Step 900..

**Our training recipe outperforms existing methods**    As demonstrated in Table 2, our novel training recipe consistently achieves superior results across all evaluated vision-centric benchmarks compared to other approaches.

**Multimodal Aha Moment**    A particularly intriguing phenomenon observed during the training of DeepSeek-R1-Zero is the occurrence of an "aha moment". This aha moment indicates DeepSeek-R1-Zero spontaneously builds a reasoning strategy, rethinking its initial approach for improved reasoning capability:

```
...
**Wait, wait. Wait. That's an aha moment I can flag here.**
Let's reevaluate this step-by-step to identify if
...
```

Following DeepSeek-R1, we extend this notion to the multimodal setting. Specifically, we define the "multimodal aha moment" we study in this work as:

> **Multimodal Aha moment** An emergent shift in multimodal large language model's reasoning behavior during RL training, where the model spontaneously begins exploring in more sophisticated, advanced strategy in visual reasoning.

Along with strong performance gain, we also observed our model exhibit very interesting "aha moment" aligned with the finding in DeepSeek R1. During training, the model spontaneously revisit its previous understanding and explore alternative options:

```
...
Therefore, dark brown wooden bed with white blanket is not above
the doorway.
**But wait! I can think of something else.**
...
```

We include more qualitative examples of this behavior in the Appendix F.

To quantitatively visualize the emergence of self-reflection behavior, we illustrate the mean and variance of frequency of reasoning-indicative keyword during training on identical samples across multiple runs in Figure 3. We observe reasoning-indicative keywords (e.g., "wait", "again") surge during Steps 600 – 900, marking the emergence of reflection and refinement strategies, and stabilize beyond Step 900, indicating consolidation of reasoning patterns.

## 5 DISCUSSION

### 5.1 INVESTIGATING THE MISSING REASONING EMERGENCE IN RL-TRAINED INSTRUCT MODELS

Starting from scratch with a non-SFT multimodal model, we have demonstrated that RL empowers **VisualThinker-R1-Zero** to attain robust reasoning capabilities without any supervised fine-tuning data. In addition to this replication, one might be inclined to apply RL directly to supervised fine-tuned models given its stronger instruction following capability. However, when attempting to replicate this success by applying RL to supervised fine-tuned models, we encountered a phenomenon: the absence of the "aha moment", where model develops emergent sophisticated reasoning behavior during training.

In following paragraphs, we share our investigations into this phenomenon, providing insights that may benefit future research. We first demonstrate how RL on instruct models leads to trivial reasoning patterns rather than genuine reasoning capabilities (Figure 5). We then share our failed attempts to elicit sophisticated reasoning from instruct models, including **freezing the vision encoder during training** and **using a length-based reward to encourage longer responses**. Despite these efforts, we were unable to induce diverse exploration of advanced reasoning behavior purely through RL training on instruct models. In contrast, base models readily exhibit such diversity, suggesting that RL on base models might be a more effective option at this stage, and leaving the challenge of eliciting similar behavior in instruct models for future work.

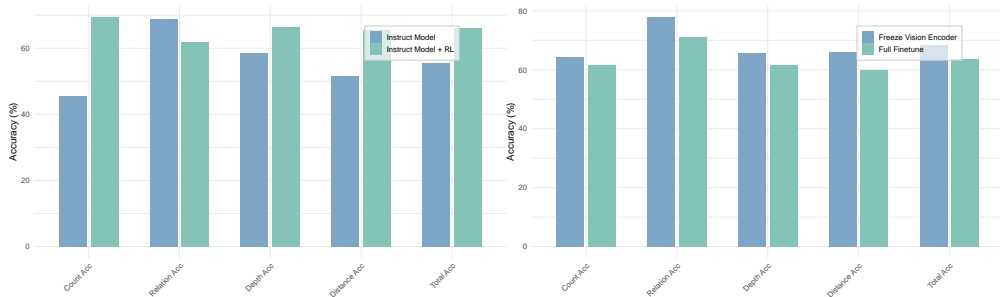

Figure 4: **Evaluation of reinforcement learning on instruct models.** (a) (left) Applying RL on instruct model indeed improves its performance. (b) (right) Freezing either the language or vision component further improves performance of RL on instruct models.

**The Missing Reasoning Patterns** Applying RL on instruct models degenerate into short and trivial reasoning rather than developing genuine reasoning capabilities (Figure 5) despite improvement on the model's performance (Figure 4 Left). The observed reasoning trajectory often follows a structure: a trivial and generic strategy within `<think></think>` tags followed by the answer between `<answer></answer>` tags. In this pattern, model bypasses the intended reasoning process and jumps directly to the final answer. While this may lead to immediate stronger training signal, it comes at the cost of exploring diverse and generalizable reasoning strategy.

---

**Trivial Reasoning Trajectory**

**Question:** Which object is closer to the camera taking this photo, the box (highlighted by a red box) or the desk (highlighted by a blue box)?
**Response:** <think>To determine which object is closer to the camera, I will measure the distance from the camera to each object. </think> <answer>box </answer>

---

Figure 5: **Example response of applying RL to supervised fine-tuned model.**

To better understand and quantify the training dynamics of instruct model, we measure the response length and entropy during training. As shown in Figure 6, the consistently higher entropy and response length of base models throughout training points to exploration of more diverse and sophisticated

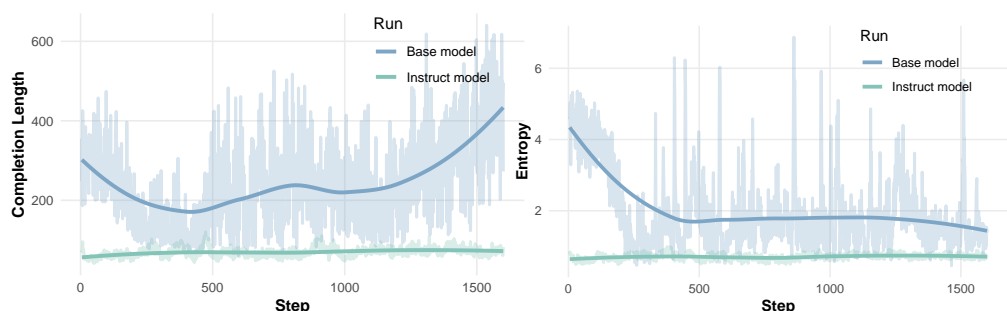

Figure 6: **RL training dynamics and example output. Left:** Completion length across training steps for RL on base and instruct models. **Right:** Entropy over tokens across training steps for RL on base and instruct models.

reasoning behaviors. In contrast, instruct models rapidly converge into trivial reasoning marked by extremely low response length and entropy, and fails to explore meaningfully throughout the training.

This finding aligns West et al. [32]'s observation that base models exhibit greater creativity than aligned models, providing a quantitative mechanism explaining why starting from base model enables more sophisticated reasoning capabilities while instruct models might remain stuck in local optima.

While instruct models may seem appealing for RL due to their stronger capabilities, we showed that it is not trivial to elicit this exploration behavior on instruct model by sharing a few of our failed attempts below.

**Can Freezing Vision Encoder Elicit "Aha Moment"?** Given the improved performance alongside the degenerating response diversity characteristic of RL training on instruct model, we hypothesize that the improvement could be attributed to improved visual representation. We thus investigated whether we could encourage reasoning exploration by freezing the visual components during training. Shown in Figure 4 and Figure 7, freezing vision components slightly improved performance, initially alleviates the decline in response length, yet it eventually leads to the same trivial reasoning patterns observed in full fine-tuning.

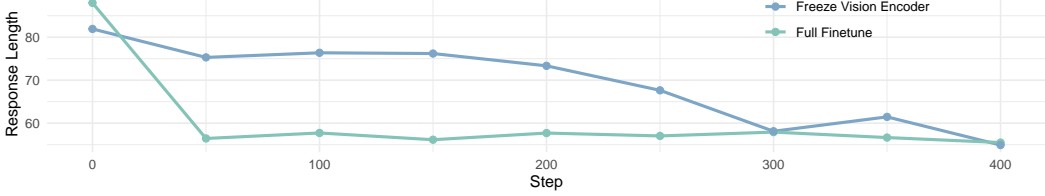

Figure 7: **Response length across training steps for different fine-tuning settings during RL.** The x-axis represents training steps, while the y-axis shows the response length. Models with different fine-tuning configurations are compared: Freeze Vision Encoder (blue), and Full Finetune (green). Despite our hypothesis, freezing the vision encoder did not force the model to develop sophisticated reasoning strategies, as evidenced by the consistently short responses length.

**Failed Attempts to Induce "Aha Moment" through Length Rewards** One key characteristic of DeepSeek R1's success is its increased response length during training. Inspired by this, we investigated whether we could encourage more sophisticated reasoning patterns by directly rewarding longer responses. Thus, we implemented a length-based reward mechanism: in addition to the vanilla accuracy and format rewards described in Section 3, we added an auxiliary reward of $+0.001$ for each additional token generated. However, as shown in Figure 8, this approach failed to produce the desired emergence of sophisticated reasoning. Instead, it led to reward hacking behaviors, with models generating extremely long yet meaningless and repetitive content as illustrated in Figure 9.

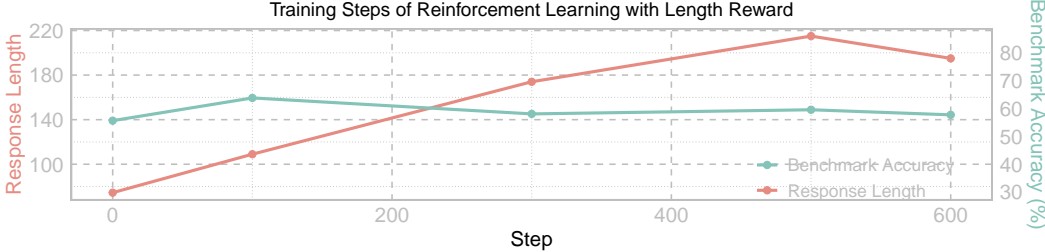

Figure 8: **Model performance during training with length-rewarded RL on instruction finetuned models.** Despite implementing length rewards to encourage longer responses, model performance shows no improvement in reasoning quality as response length increases. While average response length (red line) grows substantially, benchmark accuracy (green line) remains relatively stable, suggesting that longer responses do not necessarily translate to enhanced reasoning capabilities.

---

**Length Reward Hacking Example**

**Question:** Considering the relative positions of the giraffe (annotated by the red box) and the person in the image provided, where is the giraffe (annotated by the red box) located with respect to the person?

**Model Response:** <think> - The giraffe (annotated by the red box) is located in the foreground of the image. - The person is in the background, standing behind the giraffe. - The giraffe is not below the person. - The person is not in front of the giraffe. - ... [Repeats until it reaches max_new_tokens limit] ... </think>

---

Figure 9: **Example of reward hacking**. Model hacks reward by repeating trivial response.

## 5.2 ABLATION STUDY ON VISUALTHINKER R1

### 5.2.1 ABLATION ON COLD START

As detailed in Section 4, we stabilized the cold start training stage and unlock greater potential of our base model training with a small amount of reasoning trajectories data. To isolate the effect of cold-start trajectories, we conduct ablation studies comparing models trained solely on these trajectories versus model trained with the full recipe (i.e., VisualThinker R1 = Cold Start + GRPO). Results in Table 3 indicates that the performance gains are still primarily driven by the RL training.

Table 3: **Ablation study on the effect of cold start trajectories**. We can observe that the full training recipe is significantly higher performance comparing to model trained only on cold start data. This indicates that the performance gains are still primarily driven by the RL training.

| Training Method | Total Acc (%) | Count Acc (%) | Relation Acc (%) | Depth Acc (%) | Distance Acc (%) |
|---|---|---|---|---|---|
| Cold Start Only | 48.48 | 62.44 | 29.08 | 50.17 | 49.50 |
| Full Recipe | 70.58 | 63.32 | 79.08 | 75.00 | 66.50 |

### 5.2.2 ABLATION ON TRAINING TEMPERATURE

To understand the impact of temperature on RL training, we conduct an ablation study on varying training temperature. Overall, we observed that training with low temperature (e.g., 0.25) exhibits excessive determinism, limiting exploration and ultimately constraining the model's maximum achievable accuracy. Additionally, all settings with temperature below 1.0 showed a consistent pattern of response length shrinking in later training stages, suggestion a collapse in exploration and failure to discover more structured reasoning strategy. Thus, we adopted a training temperature of 1.0, in which we observe more well-structured reasoning.

Table 4: **Ablation study on the effect of temperature**. Training with temperature 1.0 produces model with more well-structured reasoning behavior while achieve competitive quantitative performance.

| Temperature | Total Acc (%) | Count Acc (%) | Relation Acc (%) | Depth Acc (%) | Distance Acc (%) |
|---|---|---|---|---|---|
| 0.25 | 57.16 | 59.39 | 64.77 | 55.50 | 47.67 |
| 0.50 | 61.06 | 56.59 | 71.53 | 59.00 | 57.66 |
| 0.75 | 59.51 | 62.69 | 67.53 | 53.50 | 52.66 |
| 1.00 | 59.47 | 59.64 | 66.76 | 54.16 | 56.66 |

## 6 CONCLUSION

This paper presents VisualThinker R1, the first successful multimodal replication of DeepSeek R1's emergent reasoning characteristics. By applying reinforcement learning directly to a non-fine-tuned Qwen2-VL-2B model, we observed both the "aha moment" and increased response length during training—key indicators of autonomous reasoning development. Empirically, our approach achieved 59.47% accuracy on CVBench, outperforming both base and instruction-tuned models without any supervised fine-tuning. We further incorporate a small amount of cold-start data and achieve 70.58% accuracy on CVBench, a performance surpass GPT-4o-mini. We also share our insights and failed attempts in achieving R1-like reasoning using RL with instruct model: applying RL to supervised fine-tuned models leads to trivial reasoning trajectories rather than genuine problem-solving strategies. These findings explore RL training in multimodal reasoning, potentially offering a more scalable approach to developing AI systems with robust visual reasoning capabilities.

## 7 REPRODUCIBILITY STATEMENT

We have taken several steps to ensure the reproducibility of our results. A detailed description of experimental details, including all models, packages used, and hyperparameter settings are outlined in B. Furthermore, we provide our source code in the supplementary materials.

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

CONTENTS

## A    LIMITATIONS AND FUTURE WORKS

While our study demonstrates the emergence of complex reasoning behaviors in a non-SFT 2B multimodal model, several limitations should be noted.

**Intensive Resource Demand**    One limitation of our approach is its high demand for computational resources. The adopted GRPO training approach in our work is highly resource-intensive, requiring substantial GPU memory and compute time. This is largely due to the need to generate and evaluate long CoT rollouts during training. The reasoning model often produces lengthier responses than non-reasoning models at inference time, resulting in increased decoding cost. These resource demands may pose challenges to the scalability of the approach.

**Domain and Task**    Our current exploration focuses on benchmarks that test spatial and visual reasoning capabilities but do not cover more advanced visual understanding. Specifically, the datasets we use, such as SAT and CVBench, do not require the model to interpret complex visual scenes, or focus on fine-grained visual details. Understanding how rl-based visual reasoning model generalizes to more complex visual settings remains an important direction for future exploration.

**Modality Limitations on Reasoning Steps**    Our exploration purely focuses on models reason in language. However, in cases such as describing relative spatial layouts, geometric transformations, or occlusion relationships, where text-only reasoning could be cumbersome or ambiguous, reasoning on more modalities (sketches, spatial diagrams, or visual annotations) are potentially more effective. Expanding beyond text-only reasoning remains a promising direction.

**Generalization to Other Base Models Remains Unverified**    A stronger demonstration of our conclusion's generalizability would be to apply RL directly to other base models. Unfortunately, there are very few open-source small multimodal non-sft models available, and the ones that exist are very limited in capability. While we attempted to train other open-source base models (e.g., SmolVLM [14], Paligemma2 [25], and InternVL3 [38]), we found that many of these models struggled to produce even the minimally structured outputs needed for reward supervision, particularly under the constraints of non-SFT initialization and small model size. This appears to stem not from a limitation of the RL method itself, but rather from the models' insufficient capacity to generate structured responses that can be meaningfully guided by a reward signal.

Despite these limitations, our results offer a strong proof of concept: with appropriate reward structures and training strategies, a 2B non-SFT multimodal model can develop emergent reasoning behaviors previously thought to require much larger or instruction-tuned models. While further investigation is needed, we see no clear evidence of fundamental barriers that would prevent extending this paradigm to larger models, more complex tasks, or richer multimodal interactions in the future.

## B    EXPERIMENTS SETTINGS

### B.1    IMPLEMENTATION DETAILS

All experiments are conducted using four NVIDIA H100 GPUs (80GB each), setting the batch size to 1 per device. The model is trained for $1,500$ steps with a learning rate of $1 \times 10^{-6}$ and a temperature of $1.0$. We found that allowing long responses is essential for observing increasing response length during training, so we set the maximum response length to $700$. During GRPO optimization, we sample 8 responses per step and apply a KL coefficient of $0.04$.

### B.2    EVALUATION SETTINGS

For models whose benchmark results are not publicly available, we conduct our own evaluation. We use the official evaluation code provided by the model authors if it is available. Otherwise, we evaluate the models under both the reasoning and no-reasoning settings using the corresponding prompt templates demonstrated in Figure 10. Reasoning models are evaluated with reasoning prompt template, while non-reasoning models are evaluated with non-reasoning prompt template.

Table 5: **Hyper-parameters of VisualThinker R1 GRPO training.**

| Setting | Value |
|---|---|
| Batch Size per Device | 1 |
| Gradient Accumulation Steps | 2 |
| Training Steps | 1500 |
| Learning Rate | $1 \times 10^{-6}$ |
| Temperature | 1.0 |
| Maximum Response Length | 700 |
| Number of Responses per GRPO Step | 8 |
| KL Coefficient | 0.04 |

---

**Evaluation Prompt**

**reasoning prompt:** {Question} Output the thinking process in <think> </think> and final answer in <answer> </answer> tags.
**non-reasoning prompt:** {Question}

---

Figure 10: The evaluation prompt template

### B.3 SPECIFICATION OF MODELS FOR COMPARISON

Table 6 lists the specific versions of the proprietary models used in our comparison. For GPT-4o and GPT-4o-mini, we report results based on the versions released on `2024-08-06` and `2024-07-18`, respectively.

Table 6: **The versions of proprietary model used for comparison**

| Model | Version |
|---|---|
| GPT-4o | gpt-4o-2024-08-06 |
| GPT-4o-mini | gpt-4o-mini-2024-07-18 |

### B.4 COMPUTATION RESOURCE

All experiments are conducted using four NVIDIA H100 GPUs (80GB each), setting the batch size to 1 per device. The model is trained for $1,500$ steps and takes roughly 30 hours to complete.

## C CORRELATION BETWEEN EMERGENT "AHA MOMENT" DYNAMICS AND PERFORMANCE IMPROVEMENTS

This section provides extended analysis illustrating the link between the emergence of reflective reasoning behaviors (the "aha moment") and performance improvements during reinforcement learning. Our experiments suggest a strong and highly consistent correlation pattern across multiple runs.

Across multiple RL runs of VisualThinker-7B, we consistently observe the same three-phase response-length dynamics, which aligns closely with changes in model accuracy.

**Phase 1 — Response-Length Drop (Format Compliance Phase)**: First, at the start of training, response length decreases as the model focuses on optimizing format reward. During this stage, accuracy remains stable or increases slightly.

**Phase 2 — Rapid Response-Length Growth (Emergence Phase / "Aha Moment")**:Second, a rapid emergence phase follows, characterized by a sharp increase in response length and the appearance of reflective behaviors (e.g., "wait", "let me reconsider"), as also supported by the keyword statistics shown in Table 12.

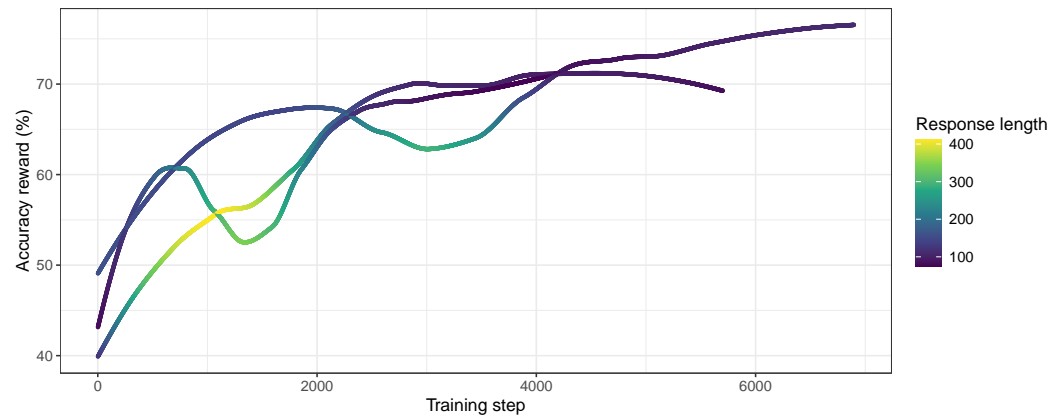

Figure 11: Across all runs, accuracy improvements consistently follow a shared sequence with the accuracy rise always aligning with the end of the emergence phase—strongly indicating that reflective-behavior emergence is tightly correlated with the development of more effective reasoning strategies under RL.

Accuracy during this stage becomes worse, consistent with increased exploratory behavior.

**Phase 3 — Stabilization After the Emergence**: Third, once this exploratory phase concludes, the model converges to a more efficient reasoning strategy, and response length stabilizes or decreases.**Importantly, it is during this post-emergence stabilization phase that accuracy consistently rises significantly across all runs.**

Across all runs, we observe a consistent ordering of events: (1) response-length drop, (2) emergence phase, (3) stabilization, and (4) accuracy improvement. The timing of the accuracy increase aligns closely with the conclusion of the emergence phase in every run. This pattern persist consistently across all runs, providing a strong empirical evidence that the emergence of reflective behavior is tightly correlated with the acquisition of more effective reasoning strategies under RL.

## D  STATISTICAL ROBUSTNESS AND MULTI-SEED EVALUATION

This section provides additional statistical analysis to ensure that the empirical trends reported in the main paper are stable, reproducible, and not dependent on a single random seed. All experiments have been re-run with three or more randomized seeds, and updated figures and tables now report mean values together with variability measures.

### D.1  CLARIFICATION OF REPORTED STATISTICS

All plots now display the mean across multiple runs, with shaded regions representing the standard error of the mean. Tables throughout the paper have been updated to report mean $\pm$ SEM wherever applicable. These additions remove ambiguity regarding whether values represent mean or median statistics and ensure that all main results are aggregated across multiple seeds rather than relying on a single initialization.

### D.2  MULTI-SEED ACCURACY RESULTS

Table 7 summarizes the multi-run performance on downstream benchmarks. The improvements remain consistent across seeds, confirming the stability of the observed gains.

### D.3  MULTI-SEED REFLECTIVE-REASONING KEYWORD DYNAMICS

To verify that the reflective-reasoning indicators are not artifacts of a particular seed, we aggregate keyword statistics across multiple runs. Table 8 reflects the updated multi-seed values corresponding to Figure 3 in the main paper.

Table 7: Multi-run evaluation on downstream benchmarks.

| Benchmark | CVBench | BLINK |
|---|---|---|
| VisualThinker-R1-Zero (Ours) | $60.24 \pm 3.40$ | $55.99 \pm 3.06$ |

Table 8: Multi-run reflective-reasoning keyword dynamics across training steps.

| Keyword | 300 | 600 | 900 | 1200 | 1500 | 1800 |
|---|---|---|---|---|---|---|
| again | $54.80 \pm 3.37$ | $71.20 \pm 29.43$ | $93.00 \pm 36.65$ | $76.20 \pm 21.27$ | $66.20 \pm 19.32$ | $72.40 \pm 28.27$ |
| wait | $3.00 \pm 1.26$ | $5.00 \pm 3.41$ | $29.80 \pm 39.34$ | $43.00 \pm 72.50$ | $62.80 \pm 50.47$ | $83.80 \pm 45.23$ |
| but | $320.20 \pm 44.24$ | $403.20 \pm 191.77$ | $386.80 \pm 167.49$ | $470.60 \pm 249.14$ | $481.20 \pm 231.87$ | $577.00 \pm 272.22$ |
| however | $171.00 \pm 30.93$ | $131.00 \pm 88.56$ | $94.60 \pm 22.52$ | $106.60 \pm 20.44$ | $73.00 \pm 20.75$ | $80.80 \pm 58.66$ |
| hmm | $1.80 \pm 1.60$ | $1.20 \pm 1.17$ | $2.40 \pm 3.32$ | $4.00 \pm 4.77$ | $5.20 \pm 2.40$ | $8.80 \pm 5.46$ |
| alternatively | $1.40 \pm 0.49$ | $1.00 \pm 1.10$ | $2.40 \pm 3.83$ | $4.00 \pm 3.03$ | $4.80 \pm 3.54$ | $4.80 \pm 4.45$ |
| check | $65.00 \pm 10.92$ | $121.00 \pm 41.51$ | $93.80 \pm 58.63$ | $120.20 \pm 124.83$ | $145.00 \pm 125.12$ | $184.60 \pm 99.69$ |

These aggregated results confirm that the qualitative trends highlighted in the main paper—response-length dynamics, emergent self-reflective phrasing, and downstream benchmark gains—hold consistently across different seeds, demonstrating that the findings are not attributable to a single lucky initialization.

## E    EXTENDED EVALUATION ON LARGER AND DIVERSE NON-SFT BASE MODELS

To assess the generality of the proposed RLVR training recipe, we extend our study beyond the 2B non-SFT base model used in the main paper. In particular, we evaluate two additional non-SFT base models: (1) **Qwen2-VL-7B**, a larger model from the same family, and (2) **InternVL-2.5-1B**, a smaller model from a different architecture family.

To ensure fair comparisons, we evaluate only the *non-SFT base variants* of these stronger model families, avoiding confounding factors introduced by large-scale instruction tuning pipelines. This isolates the contribution of our post-training recipe rather than the influence of proprietary or disproportionately large SFT data.

Across both models, we consistently observe:

1. **Accuracy gains** on CVBench,
2. **First-drop–then-rise response-length dynamics** (the characteristic "aha moment" trajectory),
3. **Reflection-indicative keyword patterns** following the same drop–rise–drop trend.

These findings demonstrate that the emergent behaviors induced by RLVR are reproducible, scalable, and architecture-agnostic.

Table 9: CVBench accuracy improvements across additional non-SFT base models.

| Model | Base Acc. | + RLVR Recipe |
|---|---|---|
| Qwen2-VL-7B | $66.22 \pm 0.00$ | $78.02 \pm 0.25$ |
| InternVL-2.5-1B | $41.88 \pm 0.00$ | $50.88 \pm 0.38$ |

Table 10: Response-length statistics for Qwen2-VL-7B across training steps.

| Steps | 0 | 300 | 600 | 900 | 1200 | 1500 |
|---|---|---|---|---|---|---|
| Length | 169.25 | 110.46 | 209.57 | 139.01 | 334.91 | 188.91 |

Table 11: Response-length statistics for InternVL-2.5-1B across training steps.

| Steps | 0 | 1200 | 2400 | 3600 | 4800 | 6000 |
|---|---|---|---|---|---|---|
| Length | 95.13 | 91.33 | 102.09 | 108.75 | 123.71 | 135.43 |

Table 12: Keyword statistics for Qwen2-VL-7B.

| Keyword / Interval | 300 | 600 | 900 | 1200 | 1500 |
|---|---|---|---|---|---|
| again | 44 | 53 | 61 | 88 | 86 |
| wait | 0 | 12 | 63 | 75 | 65 |
| but | 215 | 423 | 526 | 597 | 524 |
| however | 116 | 106 | 88 | 97 | 97 |
| hmm | 0 | 1 | 9 | 7 | 6 |
| alternatively | 0 | 3 | 3 | 4 | 4 |
| check | 295 | 288 | 296 | 309 | 295 |

Table 13: Keyword statistics for InternVL-2.5-1B.

| Keyword / Interval | 1200 | 2400 | 3600 | 4800 | 6000 |
|---|---|---|---|---|---|
| again | 67 | 185 | 205 | 189 | 62 |
| wait | 31 | 53 | 106 | 227 | 3 |
| but | 3583 | 6197 | 5787 | 5100 | 3246 |
| however | 1719 | 3213 | 3188 | 2597 | 1477 |
| hmm | 7 | 9 | 23 | 32 | 0 |
| check | 245 | 460 | 453 | 419 | 302 |
| alternatively | 21 | 36 | 41 | 49 | 3 |

## F  VISUAL EXAMPLES

Below we demonstrate qualitative examples of reasoning during training and evaluation.

**Example of multi-step reasoning during training**

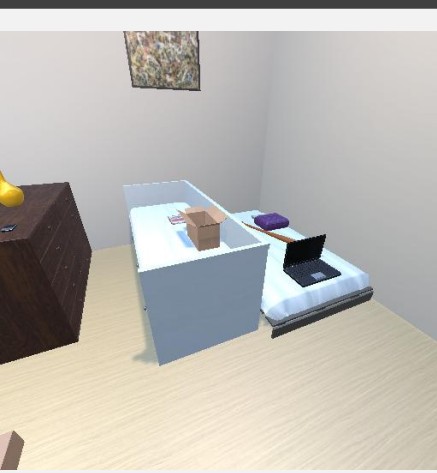

**Question:** How many Dressers are visible in the scene? Choose between the following options: 3, 0, 4, 1 or 2

**Ground truth:** 1

**Response:** Step 1: Identify the dresser in the room In the image, I can see a dresser on the left side. Step 2: Count the number of dressers The dresser is the only piece of furniture with the shape of a dresser in the image.
Conclusion: There is 1 dresser in the image. So, the answer is 1 dresser.</think> <answer>1</answer>

**Example of thinking about alternatives on CVBench**

**Question:** Estimate the real-world distances between objects in this image. Which object is closer to the books (highlighted by a red box), the lamp (highlighted by a blue box) or the chair (highlighted by a green box)? (A) lamp (B) chair

**Ground Truth:** (A) lamp

**Response:** First, I will look at the positions of the objects highlighted by the boxes. The lamp is on the right side of the image, while the chair is on the left. The books are located in the middle of the image, so they are closer to the center than either the lamp or the chair.
Next, I will consider the size of the objects. The lamp is relatively small and placed on top of the bookshelf, while the chair is larger and placed on the floor. The books are also relatively small and placed on the stool.
**Finally, I will take into account the perspective of the image. The camera is positioned at a higher angle, so the objects appear smaller and farther away than they actually are.** However, the books are still closer to the center of the image than either the lamp or the chair. Based on these observations, I conclude that the books are closer to the lamp than to the chair.</think>
So, the answer is (A) lamp. <answer>lamp</answer>,

**Example of multi-step inconsistent reasoning during training**

**Question:** Considering the relative positions, where is framed monochromatic wall decor painting (highlighted by a blue box) with respect to sofa (highlighted by a red box)? Choose between the following options: above or below

**Ground truth:** above

**Response:** The user asks for the position of the framed monochromatic wall decor painting with respect to the sofa. To answer this question, I need to know the location of both the sofa and the framed monochromatic wall decor painting. However, the location of the two objects is not provided. </think> <answer> above </answer> <reason> Since the location of both the sofa and the framed monochromatic wall decor painting is not provided, I cannot provide a detailed reasoning. However, I can provide an answer based on the information given: </reason> <answer> above </answer>

## G  ADDITIONAL ANALYSIS OF REFLECTION-INDICATIVE KEYWORDS ACROSS BASELINES

**Keyword Statistics for Baseline VLMs.**  Figure 3 in the main paper visualizes how reflection-indicative tokens (e.g., "wait," "again") dynamically emerge throughout RL training. Since this signal is collected *only during training rollouts*, equivalent mid-training curves cannot be constructed for public baseline models where intermediate checkpoints are not available. To provide a fair comparison, we evaluate each baseline model on the *same SAT prompts used in our RL rollouts* and report the keyword frequencies aggregated over fixed step intervals. Although these static statistics cannot reproduce the temporal emergence pattern seen in Figure 3, they reveal the extent to which different models naturally produce reflection-like reasoning behaviors. Among the four evaluated baselines, only **OpenVLThinker-3B** exhibits reflection keyword frequencies comparable to our model. The remaining models show minimal presence of such markers, suggesting limited spontaneous reflective reasoning. Below we present the full quantitative results (Tables 14–17), as well as the corresponding plots in Figures 12–15.

## H  USE OF LARGE LANGUAGE MODELS

We used Claude (Anthropic)[1] and ChatGPT (OpenAI)[2] as general-purpose editing tools for minor polishing of the manuscript. Specifically, the LLMs were used to:

---

[1]https://claude.ai/

[2]https://openai.com/

Table 14: Keywords statistics of **R1-VL 2B** [36].

| Interval | again | wait | but | however | hmm | alternatively | check |
|---|---|---|---|---|---|---|---|
| 0–299 | 1 | 0 | 6 | 1 | 0 | 0 | 22 |
| 300–599 | 4 | 0 | 5 | 0 | 0 | 0 | 30 |
| 600–899 | 3 | 0 | 8 | 0 | 0 | 0 | 19 |
| 900–1199 | 2 | 0 | 5 | 1 | 0 | 0 | 26 |
| 1200–1499 | 3 | 0 | 12 | 0 | 0 | 0 | 23 |

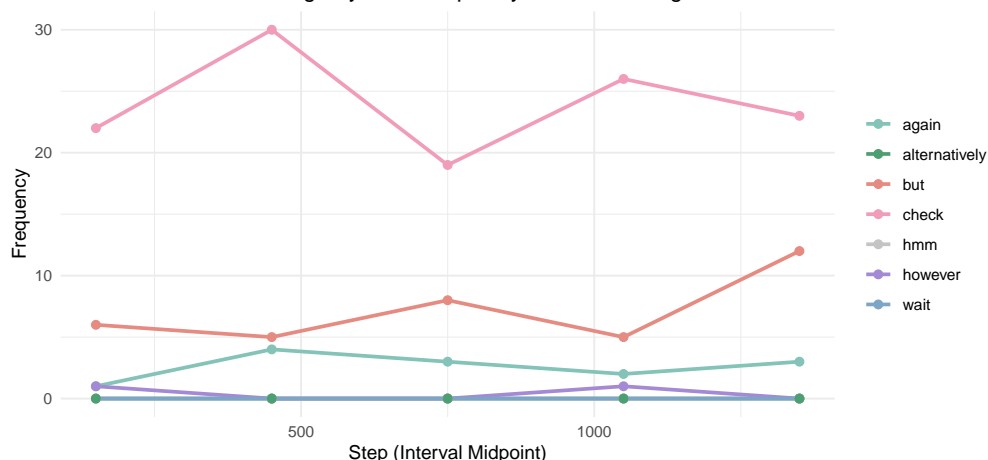

Figure 12: **Statistical analysis of reasoning-indicative keywords ("wait," "again," "but," "however," "so," "hmm," "check," "alternatively," "mistake") across evaluation intervals for baseline model R1-VL-2B [36].** The distribution highlights the degree to which reflection and refinement behaviors appear without RL fine-tuning.

- Improve grammar and sentence structure
- Enhance clarity of expression in selected passages
- Check for consistency and flow in expression

The LLMs did not contribute to research ideation, experimental design, analysis, or the generation of scientific content. All scientific claims, methodology, and core arguments are the authors' original work. The authors take full responsibility for all content in this submission.

Table 15: Keywords statistics of **VLAA-Thinker-Qwen2VL-2B** [1].

| Interval | again | wait | but | however | hmm | alternatively | check |
|----------|-------|------|-----|---------|-----|---------------|-------|
| 0–299 | 0 | 0 | 9 | 3 | 0 | 0 | 86 |
| 300–599 | 0 | 0 | 9 | 0 | 0 | 0 | 80 |
| 600–899 | 0 | 0 | 7 | 2 | 0 | 0 | 73 |
| 900–1199 | 0 | 0 | 5 | 1 | 0 | 0 | 84 |
| 1200–1499 | 0 | 0 | 2 | 0 | 0 | 0 | 86 |

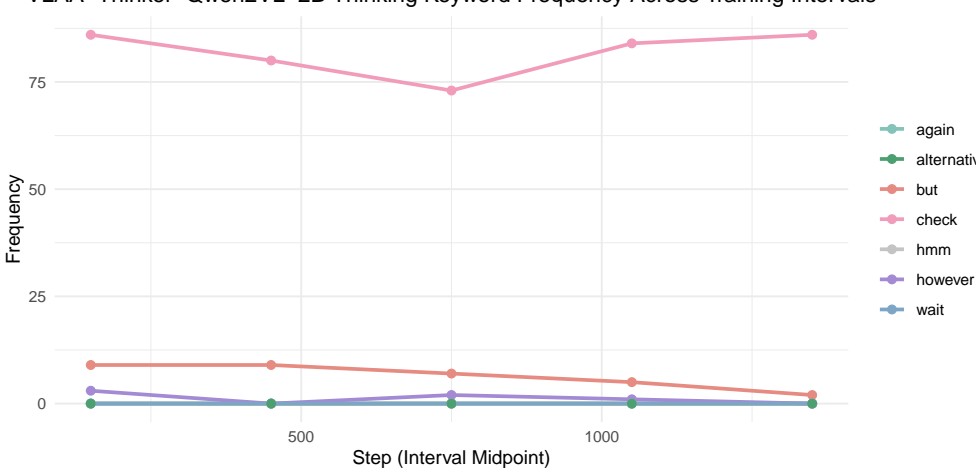

Figure 13: **Statistical analysis of reasoning-indicative keywords ("wait," "again," "but," "however," "so," "hmm," "check," "alternatively," "mistake") across evaluation intervals for baseline model VLAA-Thinker-Qwen2VL-2B [1].** The distribution highlights the degree to which reflection and refinement behaviors appear without RL fine-tuning.

Table 16: Keywords statistics of **3B-Curr-ReFT** [6].

| Interval | again | wait | but | however | hmm | alternatively | check |
|----------|-------|------|-----|---------|-----|---------------|-------|
| 0–299 | 0 | 0 | 0 | 0 | 0 | 0 | 0 |
| 300–599 | 0 | 0 | 0 | 0 | 0 | 0 | 0 |
| 600–899 | 0 | 0 | 0 | 0 | 0 | 0 | 0 |
| 900–1199 | 0 | 0 | 1 | 0 | 0 | 0 | 0 |
| 1200–1499 | 0 | 0 | 2 | 0 | 0 | 0 | 0 |

Table 17: Keywords statistics of **OpenVLThinker-3B** [7].

| Interval | again | wait | but | however | hmm | alternatively | check |
|----------|-------|------|-----|---------|-----|---------------|-------|
| 0–299 | 27 | 1 | 264 | 12 | 35 | 0 | 27 |
| 300–599 | 24 | 0 | 255 | 8 | 23 | 0 | 24 |
| 600–899 | 21 | 1 | 335 | 11 | 28 | 0 | 34 |
| 900–1199 | 26 | 1 | 320 | 14 | 32 | 0 | 18 |
| 1200–1499 | 25 | 2 | 298 | 10 | 28 | 0 | 24 |

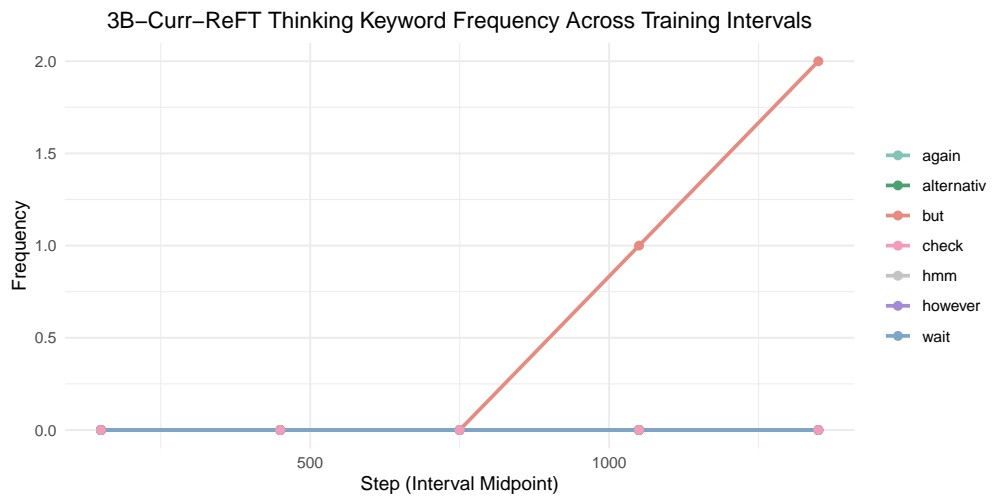

Figure 14: **Statistical analysis of reasoning-indicative keywords ("wait," "again," "but," "however," "so," "hmm," "check," "alternatively," "mistake") across evaluation intervals for baseline model 3B-Curr-ReFT [6].** The distribution highlights the degree to which reflection and refinement behaviors appear without RL fine-tuning.

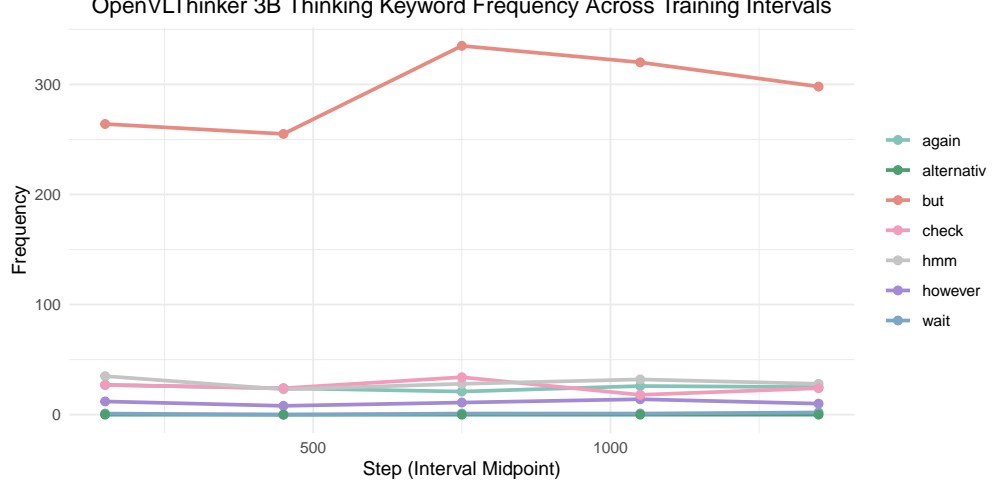

Figure 15: **Statistical analysis of reasoning-indicative keywords ("wait," "again," "but," "however," "so," "hmm," "check," "alternatively," "mistake") across evaluation intervals for baseline model OpenVLThinker-3B [7].** The distribution highlights the degree to which reflection and refinement behaviors appear without RL fine-tuning.

