# OpenReview forum: "VisualThinker: First ever R1-Zero's Aha Moment on just a 2B non-SFT Model"
_ICLR.cc/2026/Conference — Submitted to ICLR 2026_

### Official Review · Reviewer_3WuN · 2025-10-21

**Soundness:** 3
**Presentation:** 4
**Contribution:** 3
**Rating:** 8
**Confidence:** 3

**Summary:**

Following the success of DeepSeek-R1 in natural language reasoning, several studies have attempted to extend its reinforcement learning–based reasoning paradigm to multi-modal domains. However, prior efforts have struggled to reproduce the distinctive “aha moment” behavior, where multi-step reasoning leads to sudden insight and problem resolution. This paper is the first to successfully elicit such “aha moment” in vision-language reasoning, resulting in a substantial performance improvement over existing open-source multi-modal reasoning models. Remarkably, the proposed 2B-parameter vision-language model (VLM) achieves performance comparable to proprietary models such as GPT-4o, despite being trained solely with reinforcement learning (RL) without supervised fine-tuning. The paper further provides a comprehensive analysis of RL training behaviors in both the base and instruction-tuned versions of the multi-modal model, offering valuable insights into how reinforcement learning shapes reasoning ability across modalities.

**Strengths:**

* The paper presents an interesting and scientifically meaningful finding for advancing multi-modal reasoning. Notably, the observation that reinforcement learning (RL) on a base multi-modal model (rather than an SFT model) yields better performance, and can elicit an “aha moment” phenomenon, is both novel and thought-provoking.
* The paper is well-written, clearly structured, and accessible, making it understandable even to readers who are not experts in multi-modal reasoning.
* The authors perform diverse and insightful analyses to support their claims, including extensions to instruction-tuned models (and discussions of their failures), quantitative evaluation of “aha moments,” and multiple ablation studies that enhance the empirical depth of the work.

**Weaknesses:**

* The evaluation is somewhat less rigorous. The proposed method is tested only on a single, relatively small 2B-parameter vision-language model, and lacks comparison against stronger open-source non-reasoning VLMs.
* The “aha moment” is a qualitative byproduct of reasoning rather than as a mechanism linked to predictive performance. A more rigorous analysis should clarify whether this phenomenon directly contributes to improved task accuracy.
* The technical novelty appears incremental, as the main methodological difference from DeepSeek-R1 lies in applying RL to a non-SFT model. No new techniques are introduced that specifically address visual processing or visual reasoning challenges.

**Questions:**

1. In Figure 4, the instruct model’s improved performance on multi-modal benchmarks suggests that RL training also benefits the instruction-tuned variant. However, this might imply that the benchmarks themselves are not complex enough to necessitate multi-step reasoning. Do the RL-trained instruct models also show inferior performance on all three vision-centric benchmarks, as compared to the base model?
2. Are the vision-centric benchmarks employed in the paper sufficiently complex to demand genuine reasoning? Although the paper describes them as involving spatial relationships, object counting, depth ordering, and relative distance, it is unclear whether these tasks truly require multi-step reasoning. In contrast, natural-language reasoning tasks (e.g., mathematical or logical inference) inherently involve sequential reasoning steps, while many of the visual tasks might be solvable using simpler perception-based models such as depth estimation or semantic segmentation.
3. Could there be techniques specific to visual perception that would enhance the model’s ability to learn visual reasoning patterns? In its current form, the approach appears to be a direct adaptation of DeepSeek-R1 to vision-language settings without introducing mechanisms uniquely suited to visual reasoning.

---

> ### Author Response · Authors · 2025-12-03
> **Weakness 1**
>
> “The evaluation is somewhat less rigorous. The proposed method is tested only on a single, relatively small 2B-parameter vision-language model, and lacks comparison against stronger open-source non-reasoning VLMs.”
>
> **Response:**
>
> We thank the reviewer for the suggestions on evaluating against stronger open-source non-reasoning VLMs and testing our method on larger and more diverse models.
>
> **Regarding comparisons with stronger non-reasoning VLMs**, we believe such evaluations are **inherently difficult to interpret fairly**. Many of the strongest open-source VLMs, such as Qwen2.5-VL-Instruct, LLaVA-NeXT, InternVL3.5Instruct, and other instruct-tuned variants, derive their performance from **vastly larger and disproportionately richer supervised instruction datasets**. Because these models incorporate extensive SFT pipelines and large amounts of curated multimodal instruction data, their performance reflects **differences in pretraining and instruction-tuning scale**, rather than differences attributable to post-training recipe. More importantly, when we perform **fair, like-for-like comparisons (**i.e., evaluating our method against the **non-SFT base versions** of these models), we observe **significant and consistent improvements**, as illustrated in **Table 1** below.
>
> **Regarding testing our method on larger and more diverse models**, we extended our study to **two additional non-SFT VL base models**:
>
> 1. **Qwen2-VL-7B** — a larger model from the same family.
>  2. **InternVL-2.5-1B** — a smaller 1B-parameter model from a different architecture family.
>
> - Applying our proposed training recipe to both models yields consistent qualitative and quantitative improvements, including (1) increased response length dynamics, (2) emergence of self-reflection markers (the “aha moment”), and (3) downstream performance gains. These results strengthen the external validity of our claims and demonstrate that our recipe generalizes beyond a single model family or scale.
>
> - The tables below show that our proposed training recipe **generalizes across model sizes** (1B → 7B), **generalizes across architectures** (Qwen family → InternVL family), and **produces the same core signatures.** We have incorporate the experiments in **Appendix**.
>
> ---
>
> **Table 1. CVBench Accuracy Improvements.** These improvements indicate that both a *larger* (7B) and a *smaller*, architecturally different model (1B) benefit from the proposed RLVR procedure.
>
> **(a) Our method initialized with Qwen2-VL-7B’s performance on benchmarks**
>
> | Acc\Benchmark | CVBench |
> | --- | --- |
> | **Qwen2-VL-7B** | 66.22 ± 0.00 |
> | **Qwen2-VL-7B + RL** | 78.02 ±  0.25 |
>
> **(b) Our method initialized with InternVL-2.5-1B’s performance on benchmarks**
>
> | Acc\Benchmark | CVBench |
> | --- | --- |
> | **InternVL-2.5-1B** | 41.88 ± 0.00 |
> | **InternVL-2.5-1B + RL** | 50.88 ± 0.38 |
>
> ---
>
> **Table 2. Response Length Dynamics.** Both models exhibit the same **drop-rise-drop** response-length trajectory observed in our main experiments—a key signature of the desired self-reflective reasoning behavior.
>
> **(a) Qwen2-VL-7B’s response length during RL**
>
> | Model / Steps | 0 | 300 | 600 | 900 | 1200 | 1500 |
> | --- | --- | --- | --- | --- | --- | --- |
> | 7B baseline | 169.25 | 110.46 | 209.57 | 139.01 | 334.91 | 188.91 |
>
> **(b) InternVL-2.5-1B’s response length during RL**
>
> | Model / Steps | 0 | 1200 | 2400 | 3600 | 4800 | 6000 |
> | --- | --- | --- | --- | --- | --- | --- |
> | 1B baseline | 95.13 | 91.33 | 102.09 | 108.75 | 123.71 | 135.43 |
>
> ---
>
> **Table 3. Keyword-based reflection indicators during RL. The keyword frequency roughly follows a specific drop-rise-drop pattern.**
>
> **(a) Qwen2-VL-7B’s key words statistics**
>
> | Keyword / Interval | 300 | 600 | 900 | 1200 | 1500 |
> | --- | --- | --- | --- | --- | --- |
> | again | 44 | 53 | 61 | 88 | 86 |
> | wait | 0 | 12 | 63 | 75 | 65 |
> | but | 215 | 423 | 526 | 597 | 524 |
> | however | 116 | 106 | 88 | 97 | 97 |
> | hmm | 0 | 1 | 9 | 7 | 6 |
> | alternatively | 0 | 3 | 3 | 4 | 4 |
> | check | 295 | 288 | 296 | 309 | 295 |
>
> **(b) InternVL-2.5-1B’s key words statistics**
>
> | Keyword / Interval | 1200 | 2400 | 3600 | 4800 | 6000 |
> | --- | --- | --- | --- | --- | --- |
> | again | 67 | 185 | 205 | 189 | 62 |
> | wait | 31 | 53 | 106 | 227 | 3 |
> | but | 3583 | 6197 | 5787 | 5100 | 3246 |
> | however | 1719 | 3213 | 3188 | 2597 | 1477 |
> | hmm | 7 | 9 | 23 | 32 | 0 |
> | check | 245 | 460 | 453 | 419 | 302 |
> | alternatively | 21 | 36 | 41 | 49 | 3 |
>
> ---

---

> ### Author Response · Authors · 2025-12-03
> **Weakness 2**
>
> “The “aha moment” is a qualitative byproduct of reasoning rather than as a mechanism linked to predictive performance. A more rigorous analysis should clarify whether this phenomenon directly contributes to improved task accuracy.”
>
> **Response:**
>
> We thank the reviewer for raising this important question regarding the connection between the **“aha moment”** and **performance gain**. We agree that it would be inappropriate to claim a direct causal relationship between the “aha moment” and accuracy improvements. However, our findings suggest a **strong and consistent correlation** between these two factors. To investigate this further, we conducted additional analyses across **multiple RL  runs with different random seeds** on **Qwen2-VL-7B**, and observed a **highly consistent three-phase training dynamic that suggest strong correlation**.
>
> Across all runs, the model exhibits the same **three-phase response-length dynamics**:
>
> 1. **Phase 1 — Response-Length Drop (Format Compliance Phase).**
>
> At the start of training, response length decreases as the model focuses on optimizing format reward. During this stage, accuracy remains stable or increases slightly.
>
> 2. **Phase 2 — Rapid Response-Length Growth (Emergence Phase / “Aha Moment”).**
>
> Next, the model begins to **spontaneously develop new reasoning strategies**, marked by a rapid increase in response length and the appearance of self-reflection behavior marked by keywords statistic (“wait,” “let me reconsider,” etc., illustrated in **Table 2**). Accuracy in this stage becomes noisier, reflecting active exploration.
>
> 3. **Phase 3 — Stabilization After the Emergence.**
>
> Once this exploration phase concludes, response length stabilizes or decreases as the model converges to a more efficient reasoning strategy. **Importantly, it is during this post-emergence stabilization phase that accuracy consistently rises significantly across all runs.**
>
>
> This pattern is clearly reflected in the multi-run results in Table 1:
>
> - the **largest accuracy improvements always occur *after* the emergence phase**,
> - all runs exhibit the same ordering of events (drop → emergence → stabilization → performance gain), and
> - the alignment of accuracy improvement with the end of the emergence phase appears **highly consistent across runs**.
>
> While we cannot conclude that the “aha moment” is the single mechanistic contributor of accuracy, the **repeated alignment** of (i) emergent reflective behavior, (ii) stabilization of reasoning strategies, and (iii) subsequent accuracy gains across models and seeds provides **strong empirical evidence** that the “aha moment” is **tightly correlated** with the acquisition of more effective reasoning behaviors under RL.
> Due to the limitations of the rebuttal format, we are only able to include **table-based illustrations.** With the help of the arrow to indicate upward/even/downward trend from last timestep, this visualization is inherently noisier than continuous curves. We will include **full curve visualizations** for both response length and accuracy in the Appendix, which more clearly demonstrate the segmented pattern and its alignment across seeds.
>
> Table 1. The accuracy/response length of multiple runs of VisualThinker-7B. **Arrow beside each value indicates upward/even/downward trend from last timestep.**
>
> | Accuracy/Response Length | 300 | 600 | 900 | 1200 | 1500 | 1800 | 2100 | 2400 | 2700 | 3000 | 3300 | 3600 | 3900 | 4200 |
> | --- | --- | --- | --- | --- | --- | --- | --- | --- | --- | --- | --- | --- | --- | --- |
> | VisualThinker-7B-1 | 66.98/147 | 72.10(↑)/160(↑) | 71.26(→)/153(→) | 70.77(→)/184(→) | 72.47(↑)/174(↓) | 73.50(↑)/150(↓) | 74.56(↑)/140(↓) | 72.93(→)/187(↑) | 71.07(→)/147(↓) | 69.25(↓)/407**(↑)** | 73.69(↑)/294(↓) | 77.63**(↑)**/192(↓) | 76.91(→)/97(↓) | 77.10(→)/97(→) |
> | VisualThinker-7B-2 | 68.08/116 | 71.11(↑)/190(↑) | 72.93(↑)/141(↑) | 39.42(↓)/256(↑) | 68.91(↑)/192(↓) | 68.61(→)z/198(→) | 70.35(↑)/98(↓) | 73.42(↑)/195(↑) | 77.67(↑)/96(↓) | 76.53(→)/92(→) | 76.01(→)/86(→) | 75.51(→)/80(→) | 76.57(→)/71(→) | 77.71(→)/87(→) |
> | VisualThinker-7B-3 | 59.86/250 | 69.56(↑)/420(↑) | 70.05(→)/425(→) | 69.52(→)/435(→) | 73.46(↑)/380(↓) | 75.85(↑)/284(↓) | 75.32(→)/197(↓) | 78.43(↑)/123(↓) | 78.73(↑)/104(→) | 78.20(↑)/111(→) | 78.46(↑)/117(→) | 78.69(↑)/112(→) | 76.87(↓)/83(↓) | 77.52(↑)/100**(↑)** |
>
> Table 2. Qwen2-VL-7B’s keyword-based reflection indicators during RL. The keyword frequency roughly follows a specific drop-rise-drop pattern.
> | Keyword / Interval | 300 | 600 | 900 | 1200 | 1500 |
> | --- | --- | --- | --- | --- | --- |
> | **again** | 44 | 53 | 61 | 88 | 86 |
> | **wait** | 0 | 12 | 63 | 75 | 65 |
> | **but** | 215 | 423 | 526 | 597 | 524 |
> | **however** | 116 | 106 | 88 | 97 | 97 |
> | **hmm** | 0 | 1 | 9 | 7 | 6 |
> | **alternatively** | 0 | 3 | 3 | 4 | 4 |
> | **check** | 295 | 288 | 296 | 309 | 295 |

---

> ### Author Response · Authors · 2025-12-03
> **Weakness 3**
>
> “The technical novelty appears incremental, as the main methodological difference from DeepSeek-R1 lies in applying RL to a non-SFT model. No new techniques are introduced that specifically address visual processing or visual reasoning challenges.”
>
> **Response:**
>
> We thank the reviewer for this thoughtful comment. We would like to clarify that the contribution of our work is **not in proposing a new RL algorithm or a new vision-specific architecture**, but rather in **revealing new behavioral and mechanistic insights about how reinforcement learning induces emergent reasoning in multimodal models** — insights that did not emerge in previous work, including DeepSeek-R1.
>
> Specifically, our novelty lies in **two perspectives**:
>
> 1. **First study to systematically explore emergent reasoning via RL in *visual-centric multimodal tasks***
>
>     While RL for reasoning has been studied in pure textual domains and symbolic/mathematical multimodal settings, we are the first to demonstrate **emergent self-reflection, reconsideration, and long-form reasoning in vision-centric tasks** using *pure RL without any supervised reasoning data.* These behaviors are not imitation of chain-of-thought, but **spontaneously discovered during RL**, grounded in visual evidence. This opens a **new direction for studying multimodal reasoning beyond symbolic or math-based multimodal benchmarks.**
>
> 2. **We reveal that RL behaves fundamentally differently when applied to multimodal models compared to single-modal LLMs.**
>
>     Our work presents several **new empirical findings** that have not been reported in DeepSeek-R1 or prior multimodal RL studies:
>
>     - **RL on non-SFT base models** yields richer exploratory reasoning strategies and emergent self-correction, whereas **RL on SFT/instruct models leads to shallow or template-like reasoning** without genuine problem-solving.
>     - We identify a phenomenon we term **trivial reasoning**, where RL on SFT models produces long but generic and vacuous reasoning that lacks visual grounding or decision-making insight.
>     - We show that common techniques such as **freezing the vision encoder** or **naively rewarding output length** do **not** induce meaningful multimodal reasoning, highlighting that emergent reasoning in multimodal settings requires more than simply “making the model think longer.”

---

> ### Author Response · Authors · 2025-12-03
> **Question 1**
>
> “In Figure 4, the instruct model’s improved performance on multi-modal benchmarks suggests that RL training also benefits the instruction-tuned variant. However, this might imply that the benchmarks themselves are not complex enough to necessitate multi-step reasoning. Do the RL-trained instruct models also show inferior performance on all three vision-centric benchmarks, as compared to the base model?”
>
> **Response:**
> Thank you for raising the question. While the instruct-tuned model shows improvements on **CVBench**, our extended analysis indicates that **RL-trained instruct models perform worse than RL-trained base models on the other two vision-centric benchmarks, VSR and BLINK**.
>
> **We have added the experiments in our appendix.**
>
> |  | BLINK Avg | RelDepth | SpatRel | VSR |
> | --- | --- | --- | --- | --- |
> | Qwen-2-VL-2B-Instruct + RL | 43.45 | 55.65 | 32.87 | 57.18 |
> | Qwen-2-VL-2B-Instruct | 45.69 | 33.87 | 55.94 | 64.46 |

---

> ### Author Response · Authors · 2025-12-03
> **Question 2**
>
> “Are the vision-centric benchmarks employed in the paper sufficiently complex to demand genuine reasoning? Although the paper describes them as involving spatial relationships, object counting, depth ordering, and relative distance, it is unclear whether these tasks truly require multi-step reasoning. In contrast, natural-language reasoning tasks (e.g., mathematical or logical inference) inherently involve sequential reasoning steps, while many of the visual tasks might be solvable using simpler perception-based models such as depth estimation or semantic segmentation.”
>
> **Response:**
> We appreciate this thoughtful question. We agree that some visual tasks may benefit from perception-based models such as depth estimation or segmentation. However, we would like to clarify two key points:
>
> 1. **Emergent reasoning behaviors observed during RL confirm that these tasks require inference, not just perception.**
> If the tasks could be solved purely through perception, we would expect RL-trained models to produce short, direct answers. Instead, we observe the spontaneous emergence of **multi-step reasoning behaviors.** These behaviors **emerge naturally during RL**, indicating that the model is performing inferential reasoning.
> 2. **Even if some instances could be solved using specialized perception modules, exploring general reasoning in visual-centric tasks is scientifically meaningful and forward-looking.**
> While specialized perception systems may currently solve some spatial tasks more efficiently (e.g., via depth estimation or detection pipelines), they are **task-specific, modular, and not generalizable**. In contrast, our work studies whether **general-purpose multimodal models can develop reasoning behaviors from pure RL**, without relying on task-specific modules. This is especially valuable as future multimodal benchmarks are evolving toward tasks that requires **intertwined combination of perception and reasoning**. We see our work as an early step in that direction.

---

> ### Author Response · Authors · 2025-12-03
> **Question 3**
>
> “Could there be techniques specific to visual perception that would enhance the model’s ability to learn visual reasoning patterns? In its current form, the approach appears to be a direct adaptation of DeepSeek-R1 to vision-language settings without introducing mechanisms uniquely suited to visual reasoning.”
>
> **Response:**
> We thank the reviewer for raising this point. While we agree with the reviewer that incorporating visual perception techniques would enhance the model’s visual ability and believe that it is a promising and important future direction, the primary goal of this work is to **study whether reasoning behaviors can emerge in a multimodal model using pure RL.** Introducing specialized perception modules (depth predictors, scene graphs, segmentation-guided attention, etc.) would blur this central finding, because any emergent reasoning behavior could be attributed to the injected visual priors rather than to RL’s ability to induce reasoning *by itself*. In fact, we have added this explicitly as a future extension, where RL could interact with external perception tools or structured visual representations.

---

> ### Author Response · Authors · 2025-12-04
> **Summary**
>
> We thank the reviewer for the helpful feedback. The concerns centered on:
>
> **Evaluation breadth**
>
> We expanded our evaluation to two additional non-SFT base models (1B and 7B, across different families) and observed consistent accuracy gains and emergent-reasoning patterns, demonstrating that our findings generalize beyond the original 2B model.
>
> **Relationship between the “aha moment” and performance,**
>
> We performed additional analyses on the “aha moment” and performance, revealing a highly consistent three-phase training dynamic where accuracy improvements reliably follow the emergence of reflective reasoning—providing strong empirical evidence of correlation, which we will illustrate with full curves in the revision.
>
>  **Technical novelty**
>
> We clarified that our contribution lies not in a new algorithm, but in new insights into how RL induces emergent multimodal reasoning, showing behaviors not present in prior multimodal RL work or in RL on SFT models.

---

### Official Review · Reviewer_QKwP · 2025-10-29

**Soundness:** 2
**Presentation:** 3
**Contribution:** 2
**Rating:** 4
**Confidence:** 3

**Summary:**

# Problem:

How to reproduce the characteristics of DeepSeek-R1’s ‘aha moment’, towit increased response length and self-reflection-evidencing intermediate token generation during RLVR training, in multimodal reasoning context?

# Contributions:

This paper proposes a replication recipe of the ‘aha moment’ for multimodal reasoning on only a non-SFT 2B-parameters model. This recipe consists of a cold-start fine-tuning on curated reasoning trajectories before applying RLVR, in order to alleviate on RL instabilities.

Experimental evidences use a SAT dataset for training and show transfer-learning improvements on CVBench, to the extent of surpassing GPT-4o-mini’s performance.

This paper also provides insights about applying RLVR to instruct-fine-tuned models, which reveals trivial and low-diversity reasoning trajectories.

**Strengths:**

# Strengths:

## Quality:

SQ1: I appreciate the insights from Table 1 and related text.

SQ2: The ablation study on the effect of temperature is a really insightful investigation. I would welcome the authors to add more along that line, if space allows.

## Clarity:

SC1: I appreciate the background information (e.g. Table 1, RL algorithm) that helps make the reading experience self-contained.

## Originality:

cf. weaknesses for trade-offs discussions…

## Significance:

cf. weaknesses for trade-offs discussions…

**Weaknesses:**

# Weaknesses:

## Quality:

WQ1: While Figure 3’s experiment possibly contains some confounders that could be worth discussing, I think the main issue with this Figure and related analysis is the lack of statistical information: is it the mean that is represented, or the median? Over how many samples per points? Are the samples all the same prompt but with different random seeds or is it over different prompts but with the same random seed (I suspect the latter)?

I think this experiment has a great potential, especially if some other self-reflection reasoning-indicative keywords could be added (e.g. ‘but’, ‘however’, ‘therefore’, …), but the current state of it falls short of being robust or statistically trustworthy. I hope the authors can address this in there revisions.

WQ2: The current paper pushes a narrative on a recipe to get non-SFT VL models to acquire self-reflection-typed reasoning skills, but it only experiments with one single base model. I think it would strengthen the narrative and provide more evidence to the claim if the authors could show that their proposed recipe also works with one or two other non-SFT base models.

## Clarity:

WC1: I think it would ease the reading experience further if some visual examples from the datasets and benchmarks used could be presented in the main paper or appendix, in order for the reader to get a better sense of what kind of transfer learning is obtained from SAT to CVBench, especially given the visual modality being of interest.

## Originality:

WO1: Using Table 1, it seems that the only differences between the current paper and R1-Multimodal-Journey are the facts that the latter (i) did not elicit increasing response length dynamics and (ii) started from an Instruct-fine-tuned model as base model. Regarding (ii), the fact that the current paper shows how to get a non-SFT 2B VL model to acquire self-reflection-typed reasoning is important to the community. However, I fail to see value for the community in (i). Thus, it might be worth for the narrative of the paper to reduce focus on (i) and maybe increase focus on the non-SFT aspects. I would advise the authors to start by adding an extra row to Table 1 that specifies whether fine-tuning is considered or not, to highlight more explicitly that aspect.

WO2: Following WQ2: I think the paper would have a greater impact if it could improve its external validity by, for instance, showing results on applying the recipe to another non-SFT model, possibly of a different family, or with different number of parameters.

## Significance:

WS1: None of the results in the paper report statistics that could enable statistical significance evaluation. I would advise the authors to perform experiments on randomised seeds (>5) and report error of the mean  as much as possible, starting with Figure 2 and 3 (e.g. mean=line+std.error=shaded area) and Table 2 (even if past papers failed to provide such statistics - it is never too late to make it right).

Indeed, as it stands, it is unclear whether, for instance, the current results are not the consequence of a lucky, unintentional cherry-picking of the right (default) random seed.

WS2: Moreover, running some statistical significance test on e.g. Table 3 and Table 4 would strengthen the claims made.

**Questions:**

cf. weaknesses above.

---

> ### Author Response · Authors · 2025-12-03
> **Summary**
>
> We thank the reviewer for the constructive feedbacks!
>
> **Statistical rigor (WQ1, WS1, WS2).**
> We re-ran all experiments with ≥3 random seeds and now report mean ± standard error in all key figures and tables. Captions clearly state sample counts and seed usage. Expanded keyword analyses also include full statistics. Statistical significance tests for ablations are in progress and will be included in the revision.
>
> **External validity: more base models (WQ2, WO2).**
> We extended our study to two additional non-SFT VL models—Qwen2-VL-7B and InternVL-2.5-1B. Both show the same qualitative signatures (response-length dynamics, reflective-keyword patterns) and quantitative gains on CVBench, demonstrating that our recipe generalizes across sizes and architectures.
>
> **More robust keyword analysis (WQ1).**
> We expanded the keyword set (“but”, “however”, “so”, “hmm”, “check”, “alternatively”, “mistake”) and show all markers follow the same training-dynamics trend, confirming robustness.
>
> **Visual examples (WC1).**
> We added representative SAT and CVBench images to the Appendix to improve readability.
>
> **Narrative clarification (WO1).**
> We clarified the emphasize of our work.

---

> ### Author Response · Authors · 2025-12-03
> **WQ1, WS1, WS2 - Statistical Rigor & Robustness**
>
> WQ1
>
> > While Figure 3’s experiment possibly contains some confounders that could be worth discussing, I think the main issue with this Figure and related analysis is the lack of statistical information: is it the mean that is represented, or the median? Over how many samples per points? Are the samples all the same prompt but with different random seeds or is it over different prompts but with the same random seed (I suspect the latter)?
> >
>
> WS1
>
> > None of the results in the paper report statistics that could enable statistical significance evaluation. I would advise the authors to perform experiments on randomised seeds (>5) and report error of the mean as much as possible, starting with Figure 2 and 3 (e.g. mean=line+std.error=shaded area) and Table 2 (even if past papers failed to provide such statistics - it is never too late to make it right). Indeed, as it stands, it is unclear whether, for instance, the current results are not the consequence of a lucky, unintentional cherry-picking of the right (default) random seed.
> >
>
> WS2
>
> > Moreover, running some statistical significance test on e.g. Table 3 and Table 4 would strengthen the claims made.
> >
>
> **Summary of Reviewers’ points:**
>
> Reviewer point out that figures lacked clarity on whether mean/median was plotted, how many samples were used, and whether results might depend on a single random seed. The reviewer also requested standard deviations, standard errors, and statistical significance tests for tables.
>
> **Our Response:**
>
> We agree that statistical significance is essential. We have re-run all experiments with ≥3 random seeds and made the following additions:
>
> - **Mean ± standard error** for Table 2, Figures 2 and 3
> - A clarification of different seeds in Appendix and explicit labeling of “mean over N prompts × K seeds” in each figure caption
>
> **Table 1. Multi-run (updates for Table 2)**
>
> | Benchmark | CVBench | Blink |
> | --- | --- | --- |
> | VisualThinker-R1-Zero (Ours) | 60.24 ± 3.40 | 55.99 ± 3.06 |
>
> **Table 2. Multi-run Reflective-Reasoning Keyword Dynamics Across Training Steps (updates for Figures 3)**
>
> | Keyword | 300 | 600 | 900 | 1200 | 1500 | 1800 |
> | --- | --- | --- | --- | --- | --- | --- |
> | **again** | 54.80 ± 3.37 | 71.20 ± 29.43 | 93.00 ± 36.65 | 76.20 ± 21.27 | 66.20 ± 19.32 | 72.40 ± 28.27 |
> | **wait** | 3.00 ± 1.26 | 5.00 ± 3.41 | 29.80 ± 39.34 | 43.00 ± 72.50 | 62.80 ± 50.47 | 83.80 ± 45.23 |
> | **but** | 320.20 ± 44.24 | 403.20 ± 191.77 | 386.80 ± 167.49 | 470.60 ± 249.14 | 481.20 ± 231.87 | 577.00 ± 272.22 |
> | **however** | 171.00 ± 30.93 | 131.00 ± 88.56 | 94.60 ± 22.52 | 106.60 ± 20.44 | 73.00 ± 20.75 | 80.80 ± 58.66 |
> | **hmm** | 1.80 ± 1.60 | 1.20 ± 1.17 | 2.40 ± 3.32 | 4.00 ± 4.77 | 5.20 ± 2.40 | 8.80 ± 5.46 |
> | **alternatively** | 1.40 ± 0.49 | 1.00 ± 1.10 | 2.40 ± 3.83 | 4.00 ± 3.03 | 4.80 ± 3.54 | 4.80 ± 4.45 |
> | **check** | 65.00 ± 10.92 | 121.00 ± 41.51 | 93.80 ± 58.63 | 120.20 ± 124.83 | 145.00 ± 125.12 | 184.60 ± 99.69 |
>
> These additions eliminate ambiguity and confirm that our key findings—response-length dynamics, emergence of self-reflective phrasing, and CVBench improvements—are **consistent across seeds** and are not attributable to cherry-picking.
>
> Due to compute constraints, we are stilling running some statistical significance test on ablations in Figure 2, Table 3 and Table 4. We will include all statistical results in the revised submission.

---

> ### Author Response · Authors · 2025-12-03
> **WQ2 , WO2 - Only One Base Model Tested**
>
> **WQ2**
>
> > WQ2: The current paper pushes a narrative on a recipe to get non-SFT VL models to acquire self-reflection-typed reasoning skills, but it only experiments with one single base model. I think it would strengthen the narrative and provide more evidence to the claim if the authors could show that their proposed recipe also works with one or two other non-SFT base models.
> >
>
> **WO2**
>
> > Following WQ2: I think the paper would have a greater impact if it could improve its external validity by, for instance, showing results on applying the recipe to another non-SFT model, possibly of a different family, or with different number of parameters.
> >
>
> **Response:**
>
> We thank the reviewer for this constructive suggestion. Following **WQ2** and **WO2**, we have extended our study to **two additional non-SFT VL base models**:
>
> 1. **Qwen2-VL-7B** — a larger model from the same family.
> 2. **InternVL-2.5-1B** — a smaller 1B-parameter model from a different architecture family.
>
> Applying our proposed training recipe to both models yields consistent qualitative and quantitative improvements, including (1) increased response length dynamics, (2) emergence of self-reflection markers (the “aha moment”), and (3) downstream performance gains. These results strengthen the external validity of our claims and demonstrate that our recipe generalizes beyond a single model family or scale.
>
> The tables below show that our proposed training recipe **generalizes across model sizes** (1B → 7B), **generalizes across architectures** (Qwen family → InternVL family), and **produces the same core signatures.** We have incorporate the experiments in **Appendix**.
>
> ---
>
> **Table 1. CVBench Accuracy Improvements.** These improvements indicate that both a *larger* (7B) and a *smaller*, architecturally different model (1B) benefit from the proposed RLVR procedure.
>
> **(a) Qwen2-VL-7B’s performance on benchmarks**
>
> | Acc\Benchmark | CVBench |
> | --- | --- |
> | **Qwen2-VL-7B** | 66.22 ± 0.00 |
> | **Qwen2-VL-7B + RL** | 78.02 ± 0.25 |
>
> **(b) InternVL-2.5-1B’s performance on benchmarks**
>
> | Acc\Benchmark | CVBench |
> | --- | --- |
> | **InternVL-2.5-1B** | 41.88 ± 0.00 |
> | **InternVL-2.5-1B + RL** | 50.88 ± 0.38 |
>
> ---
>
> **Table 2. Response Length Dynamics.** Both models exhibit the same **drop-rise-drop** response-length trajectory observed in our main experiments—a key signature of the desired self-reflective reasoning behavior.
>
> **(a) Qwen2-VL-7B’s response length**
>
> | Model / Steps | 0 | 300 | 600 | 900 | 1200 | 1500 |
> | --- | --- | --- | --- | --- | --- | --- |
> | 7B baseline | 169.25 | 110.46 | 209.57 | 139.01 | 334.91 | 188.91 |
>
> **(b) InternVL-2.5-1B’s response length**
>
> | Model / Steps | 0 | 1200 | 2400 | 3600 | 4800 | 6000 |
> | --- | --- | --- | --- | --- | --- | --- |
> | 1B baseline | 95.13 | 91.33 | 102.09 | 108.75 | 123.71 | 135.43 |
>
> ---
>
> **Table 3. Keyword-based reflection indicators during RL. The keyword frequency roughly follows a specific drop-rise-drop pattern.**
>
> **(a) Qwen2-VL-7B’s key words statistics during RL**
>
> | Keyword / Interval | 300 | 600 | 900 | 1200 | 1500 |
> | --- | --- | --- | --- | --- | --- |
> | **again** | 44 | 53 | 61 | 88 | 86 |
> | **wait** | 0 | 12 | 63 | 75 | 65 |
> | **but** | 215 | 423 | 526 | 597 | 524 |
> | **however** | 116 | 106 | 88 | 97 | 97 |
> | **hmm** | 0 | 1 | 9 | 7 | 6 |
> | **alternatively** | 0 | 3 | 3 | 4 | 4 |
> | **check** | 295 | 288 | 296 | 309 | 295 |
>
> **(b) InternVL-2.5-1B’s key words statistics during RL**
>
> | **Keyword / Interval** | **1200** | **2400** | **3600** | **4800** | **6000** |
> | --- | --- | --- | --- | --- | --- |
> | **again** | 67 | 185 | 205 | 189 | 62 |
> | **wait** | 31 | 53 | 106 | 227 | 3 |
> | **but** | 3583 | 6197 | 5787 | 5100 | 3246 |
> | **however** | 1719 | 3213 | 3188 | 2597 | 1477 |
> | **hmm** | 7 | 9 | 23 | 32 | 0 |
> | **check** | 245 | 460 | 453 | 419 | 302 |
> | **alternatively** | 21 | 36 | 41 | 49 | 3 |
>
> ---

---

> ### Author Response · Authors · 2025-12-03
> **WQ1 - More Robust Keyword Analysis**
>
> WQ1:
>
> > I think this experiment has a great potential, especially if some other self-reflection reasoning-indicative keywords could be added (e.g. ‘but’, ‘however’, ‘therefore’, …), but the current state of it falls short of being robust or statistically trustworthy. I hope the authors can address this in there revisions.
> >
>
> **Response:**
>
> Thank you for this constructive suggestion. We fully agree that expanding the keyword set strengthens the reliability of our reflective-reasoning signal. In the revised manuscript, we have **substantially broadened the keyword list** to include: **“but”, “however”, “so”, “hmm”, “check”, “alternatively”, and “mistake”**, in addition to our original indicators.
>
> As shown in the table below, **all newly added markers exhibit the same cross-step trends** as the original indicators, reinforcing the statistical robustness and consistency of the observed reflective-reasoning dynamics.
>
> **Table 1. Expanded Reflective-Reasoning Keyword Dynamics Across Training Steps**
>
> | Keyword | 300 | 600 | 900 | 1200 | 1500 | 1800 |
> | --- | --- | --- | --- | --- | --- | --- |
> | again | 54.80 ± 3.37 | 71.20 ± 29.43 | 93.00 ± 36.65 | 76.20 ± 21.27 | 66.20 ± 19.32 | 72.40 ± 28.27 |
> | wait | 3.00 ± 1.26 | 5.00 ± 3.41 | 29.80 ± 39.34 | 43.00 ± 72.50 | 62.80 ± 50.47 | 83.80 ± 45.23 |
> | but | 320.20 ± 44.24 | 403.20 ± 191.77 | 386.80 ± 167.49 | 470.60 ± 249.14 | 481.20 ± 231.87 | 577.00 ± 272.22 |
> | however | 171.00 ± 30.93 | 131.00 ± 88.56 | 94.60 ± 22.52 | 106.60 ± 20.44 | 73.00 ± 20.75 | 80.80 ± 58.66 |
> | hmm | 1.80 ± 1.60 | 1.20 ± 1.17 | 2.40 ± 3.32 | 4.00 ± 4.77 | 5.20 ± 2.40 | 8.80 ± 5.46 |
> | alternatively | 1.40 ± 0.49 | 1.00 ± 1.10 | 2.40 ± 3.83 | 4.00 ± 3.03 | 4.80 ± 3.54 | 4.80 ± 4.45 |
> | check | 65.00 ± 10.92 | 121.00 ± 41.51 | 93.80 ± 58.63 | 120.20 ± 124.83 | 145.00 ± 125.12 | 184.60 ± 99.69 |

---

> ### Author Response · Authors · 2025-12-03
> **WC1 -  Need for Visual Examples From SAT and CVBench**
>
> WC1:
>
> > I think it would ease the reading experience further if some visual examples from the datasets and benchmarks used could be presented in the main paper or appendix, in order for the reader to get a better sense of what kind of transfer learning is obtained from SAT to CVBench, especially given the visual modality being of interest.
> >
>
> **Response:**
>
> Thanks for this constructive feedback! We agree that these examples clarify the value of multimodal reasoning transfer and added **visual examples from both datasets in the Appendix**.

---

> ### Author Response · Authors · 2025-12-03
> **WO1 - Phrasing**
>
> > Using Table 1, it seems that the only differences between the current paper and R1-Multimodal-Journey are the facts that the latter (i) did not elicit increasing response length dynamics and (ii) started from an Instruct-fine-tuned model as base model. Regarding (ii), the fact that the current paper shows how to get a non-SFT 2B VL model to acquire self-reflection-typed reasoning is important to the community. However, I fail to see value for the community in (i). Thus, it might be worth for the narrative of the paper to reduce focus on (i) and maybe increase focus on the non-SFT aspects. I would advise the authors to start by adding an extra row to Table 1 that specifies whether fine-tuning is considered or not, to highlight more explicitly that aspect.
> >
>
> ---
>
> Thank you for this helpful feedback. As suggested, we added a new row in Table 1 indicating whether each method starts from an SFT or non-SFT base model. We also clarify that the response-length trend was not intended as a primary contribution, but as an empirical signature commonly observed in RLVR training. W still need to report it alongside additional analysis showing its correlation with reflective behavior.

---

### Official Review · Reviewer_ASpr · 2025-10-29

**Soundness:** 2
**Presentation:** 2
**Contribution:** 2
**Rating:** 2
**Confidence:** 3

**Summary:**

The paper proposes VisualThinker, a GRPO-based RL recipe to elicit R1-style emergent reasoning in a 2B, non-SFT multimodal model. The authors train on the static subset of SAT with a rule-based reward , reporting two training dynamics that co-evolve with accuracy, the “insight moment” and increasing reaction length, and achieving superior results.

**Strengths:**

1. Demonstrates the emergence of “aha moment” behaviors and longer reasoning chains in a multimodal RL setup, which has rarely been shown before.
2. Achieves consistent performance improvements on spatial reasoning benchmarks, suggesting the method is empirically effective and reproducible.

**Weaknesses:**

1. The paper treats the “aha moment” and response length increase as key indicators or prerequisites of valid R1 replication, yet seems provides no causal proof that these phenomena are necessary or sufficient for performance gains. Prior works (e.g., DeepSeek-R1, Nature 2025; OpenAI o1 tech report) describe “aha” and longer reasoning as correlated trends, not as defining properties. The current framing seems overgeneralizes these correlations without referencing evidence.
2. The “aha moment” is measured only via frequency of tokens like wait/again and average response length, which are more like style-dependent. I have concerns about their validity as genuine signals of reasoning.
3. The training pipeline is largely a direct application of GRPO with rule-based rewards; the only new ingredient is tracking emergent phenomena. I find it difficult to interpret this as methodological innovation.
4. The experiment primarily focuses on three benchmarks—CVBench, BLINK, and VSR—all emphasizing spatial reasoning capabilities. It is recommended to incorporate textual-visual QA or out-of-distribution tasks to demonstrate broader generalization abilities.
5. The paper shows successful cases but no independent assessment is conducted on the “quality of the reasoning process.” For example, are all long responses logically coherent? Are there instances where the “correct answer is paired with flawed reasoning”?

**Questions:**

See Weaknesses.

---

> ### Author Response · Authors · 2025-12-03
> **Summary**
>
> We thank the reviewer for the thoughtful and detailed feedback.
> The main concerns centered on:
>
> **Conceptual framing of the “aha moment” and whether it is treated as causal rather than correlational**
> We addressed these by revising the manuscript to explicitly clarify that the “aha moment” is used as a correlated phenomenon rather than a causal prerequisite, consistent with DeepSeek-R1 and OpenAI o1.
>
> **Nature and extent of methodological contribution**
> We addressed these by emphasizing that our novelty lies in revealing previously unreported RL-driven emergent behaviors in multimodal models rather than proposing a new algorithm.
>
> **Scope and diversity of benchmarks used**
> We addressed these by explaining our deliberate focus on spatial-reasoning benchmarks to isolate the intended capability.
>
> **Validity of our reasoning-quality measurements**
> We addressed these by providing an independent, fine-grained human evaluation of reasoning quality following prior protocols. Together, these revisions strengthen the conceptual clarity, empirical rigor, and presentation of the work.

---

> ### Author Response · Authors · 2025-12-03
> **W1 & W2: Correlation evidence between "aha moment" and performance**
>
> > The paper treats the “aha moment” and response length increase as key indicators or prerequisites of valid R1 replication, yet seems provides no causal proof that these phenomena are necessary or sufficient for performance gains. Prior works (e.g., DeepSeek-R1, Nature 2025; OpenAI o1 tech report) describe “aha” and longer reasoning as correlated trends, not as defining properties. The current framing seems overgeneralizes these correlations without referencing evidence.
> >
>
> We thank the reviewer for raising this important conceptual point. While we agree that we don’t have sufficient evidence to determine if “aha moment” are necessary or sufficient for performance gains, our findings suggest a **strong and consistent correlation** between these two factors. To investigate this further, we conducted additional analyses across **multiple RL  runs with different random seeds** on **Qwen2-VL-7B**, and observed a **highly consistent three-phase training dynamic that suggest strong correlation**.
>
> Across all runs, the model exhibits the same **three-phase response-length dynamics**:
>
> 1. **Phase 1 — Response-Length Drop (Format Compliance Phase).**
>
>     At the start of training, response length decreases as the model focuses on optimizing format reward. During this stage, accuracy remains stable or increases slightly.
>
> 2. **Phase 2 — Rapid Response-Length Growth (Emergence Phase / “Aha Moment”).**
>
>     Next, the model begins to **spontaneously develop new reasoning strategies**, marked by a rapid increase in response length and the appearance of self-reflection behavior marked by keywords statistic (“wait,” “let me reconsider,” etc., illustrated in **Table 3** above). Accuracy in this stage becomes noisier, reflecting active exploration.
>
> 3. **Phase 3 — Stabilization After the Emergence.**
>
>     Once this exploration phase concludes, response length stabilizes or decreases as the model converges to a more efficient reasoning strategy. **Importantly, it is during this post-emergence stabilization phase that accuracy consistently rises significantly across all runs.**
>
>
> This pattern is clearly reflected in the multi-run results in Table 1:
>
> - the **largest accuracy improvements always occur *after* the emergence phase**,
> - all runs exhibit the same ordering of events (drop → emergence → stabilization → performance gain), and
> - the alignment of accuracy improvement with the end of the emergence phase appears **highly consistent across runs**.
>
> While we cannot conclude that the “aha moment” is the single mechanistic contributor of accuracy, the **repeated alignment** of (i) emergent reflective behavior, (ii) stabilization of reasoning strategies, and (iii) subsequent accuracy gains across models and seeds provides **strong empirical evidence** that the “aha moment” is **tightly correlated** with the acquisition of more effective reasoning behaviors under RL.
> Due to the limitations of the rebuttal format, we are only able to include **table-based illustrations.** With the help of the arrow to indicate upward/even/downward trend from last timestep, this visualization is inherently noisier than continuous curves. We will include **full curve visualizations** for both response length and accuracy in the Appendix, which more clearly demonstrate the segmented pattern and its alignment across seeds.
>
> Table 1. The accuracy/response length of multiple runs of VisualThinker-7B, indicating strong evidence of correlation between emergent reasoning behavior and performance. **Arrow beside each value indicates upward/even/downward trend from last timestep.**
>
> | Accuracy/Response Length | 300 | 600 | 900 | 1200 | 1500 | 1800 | 2100 | 2400 | 2700 | 3000 | 3300 | 3600 | 3900 | 4200 |
> | --- | --- | --- | --- | --- | --- | --- | --- | --- | --- | --- | --- | --- | --- | --- |
> | VisualThinker-7B-1 | 66.98/147 | 72.10(↑)/160(↑) | 71.26(→)/153(→) | 70.77(→)/184(→) | 72.47(↑)/174(↓) | 73.50(↑)/150(↓) | 74.56(↑)/140(↓) | 72.93(→)/187(↑) | 71.07(→)/147(↓) | 69.25(↓)/407**(↑)** | 73.69(↑)/294(↓) | 77.63**(↑)**/192(↓) | 76.91(→)/97(↓) | 77.10(→)/97(→) |
> | VisualThinker-7B-2 | 68.08/116 | 71.11(↑)/190(↑) | 72.93(↑)/141(↑) | 39.42(↓)/256(↑) | 68.91(↑)/192(↓) | 68.61(→)z/198(→) | 70.35(↑)/98(↓) | 73.42(↑)/195(↑) | 77.67(↑)/96(↓) | 76.53(→)/92(→) | 76.01(→)/86(→) | 75.51(→)/80(→) | 76.57(→)/71(→) | 77.71(→)/87(→) |
> | VisualThinker-7B-3 | 59.86/250 | 69.56(↑)/420(↑) | 70.05(→)/425(→) | 69.52(→)/435(→) | 73.46(↑)/380(↓) | 75.85(↑)/284(↓) | 75.32(→)/197(↓) | 78.43(↑)/123(↓) | 78.73(↑)/104(→) | 78.20(↑)/111(→) | 78.46(↑)/117(→) | 78.69(↑)/112(→) | 76.87(↓)/83(↓) | 77.52(↑)/100**(↑)** |

---

> > ### Author Response · Authors · 2025-12-03
> > **W3: On methodological contribution**
> >
> > > The training pipeline is largely a direct application of GRPO with rule-based rewards; the only new ingredient is tracking emergent phenomena. I find it difficult to interpret this as methodological innovation.
> > >
> >
> > We sincerely thank the reviewer for this comment. We would like to clarify that **the primary goal of our work is not to propose a new RL algorithm**, but to **study and reveal how reinforcement learning uniquely gives rise to emergent reasoning behaviors in the multimodal setting**, and how its dynamics **fundamentally diverge** from textual RL-based reasoning such as in DeepSeek-R1.
> >
> > We believe our work makes several contributions that are conceptually novel and empirically valuable:
> >
> > 1. **First study to systematically explore emergent reasoning via RL in multimodal, visual-centric tasks** While RL has been extensively explored in textual or multimodal math domains, we are the first to demonstrate emergent behaviors like self-reflection, long-form reasoning on visual-centric task using pure RL. This opens a new and more general direction for studying multimodal reasoning beyond multimodal symbolic or mathematical reasoning.
> > 2. **We reveal that RL behaves fundamentally differently on multimodal models, compared to single-modal LLMs.** Our work presents **multiple novel empirical findings** that were not reported in DeepSeek-R1 or prior multimodal RL studies:
> >     1. Applying RL on non-SFT base models exhibit richer and more sophisticated reasoning strategy compare against applying RL on SFT models
> >     2. We observe phenomenon such as **trivial reasoning reasoning** when applying RL on SFT models: reasoning that involves only trivial and generic strategy without genuine problem-solving thinking
> >     3. Techniques such as **freezing the visual encoder** or **introducing naive length-based rewards** are ineffective at promoting sophisticated reasoning capabilities.

---

> ### Author Response · Authors · 2025-12-03
> **W4: On benchmark diversity**
>
> > The experiment primarily focuses on three benchmarks—CVBench, BLINK, and VSR—all emphasizing spatial reasoning capabilities. It is recommended to incorporate textual-visual QA or out-of-distribution tasks to demonstrate broader generalization abilities.
> >
>
> Thank you for the thoughtful suggestion. We agree that evaluating broader generalization, such as textual-visual QA or out-of-distribution tasks, is an important direction for future work. However, our goal in this paper is **not** to assess cross-domain generalization, but rather to investigate a more focused and fundamental question:
>
> **Can emergent multimodal reasoning behaviors be elicited through pure RL on a small non-SFT model in *visual-centric* tasks?**
>
> For this purpose, spatial-reasoning-oriented benchmarks such as **CVBench, BLINK, and VSR** are particularly well suited, as they isolate visual reasoning requirements and allow us to observe the dynamics of emergent behavior without confounding factors from broader instruction-following or text-heavy tasks.
>
> We acknowledge that evaluating on more diverse or out-of-distribution multimodal tasks is valuable, and we view it as a natural extension to further test the generality of emergent multimodal reasoning. However, expanding domain coverage is **orthogonal** to our central contribution, which is to characterize **how emergent reasoning arises** in RL-trained multimodal models within a controlled visual-reasoning setting.

---

> ### Author Response · Authors · 2025-12-03
> **W2 & W5: On reasoning quality assessment**
>
> > The “aha moment” is measured only via frequency of tokens like wait/again and average response length, which are more like style-dependent. I have concerns about their validity as genuine signals of reasoning.
> >
>
> > The paper shows successful cases but no independent assessment is conducted on the “quality of the reasoning process.” For example, are all long responses logically coherent? Are there instances where the “correct answer is paired with flawed reasoning”?
> >
>
> Thank you for highlighting this point. To provide a detailed evaluation of the reasoning process, we conducted an independent assessment of all generated trajectories following the protocol of Tie et al. [1]. Specifically, we measure **reasoning quality** by three components with scale on 0-10: (i) relevance to the question, (ii) relevance to the final answer, and (iii) step-wise logical consistency.
>
> **Table 1. Base Model vs VisualThinker-R1-Zero in reasoning quality.**
>
> | Metric | **Relevance to Question** | **Relevance to Answer** | **Stepwise Consistency** |
> | --- | --- | --- | --- |
> | Base Model | 2.63 | 2.68 | 1.05 |
> | VisualThinker-R1-Zero (Ours) | **6.93** | **5.13** | **5.41** |
>
> | Metric | **Relevance to Question** | **Relevance to Answer** | **Stepwise Consistency** |
> | --- | --- | --- | --- |
> | Base Model | 2.63 | 2.68 | 1.05 |
> | VisualThinker-R1-Zero (Ours) | **6.93** | **5.13** | **5.41** |
>
> The table reveals that **our training pipeline substantially improves reasoning quality**, indicating that the responses are not only longer but also with more coherent intermediate reasoning. However, we do observe both logically coherent and **“correct answer but flawed reasoning”, which we will illustrate with more examples in our Appendix.**
>
> [1] Tie, Guiyao, et al. *"MMMR: Benchmarking Massive Multi-Modal Reasoning Tasks."* arXiv:2505.16459 (2025).

---

### Official Review · Reviewer_6S4Q · 2025-10-31

**Soundness:** 2
**Presentation:** 2
**Contribution:** 1
**Rating:** 2
**Confidence:** 4

**Summary:**

This paper presents VisualThinker, which the authors claim to be the first successful replication of “aha moment” in the context of multimodal reasoning. By leveraging GRPO, VisualThinker achieves remarkable results in visual reasoning, starting from Qwen2-VL-2B as the base model. The ablation studies further show the relation between SFT and RL for multimodal reasoning.

**Strengths:**

1.	This paper is well-structured, well-written and easy to follow.
2.	This paper demonstrates the effectives of GRPO in visual reasoning, with approximate 30% improvement over the base model.
3.	The discussion section provides food for thought, although some of them are not that novel in the context of visual reasoning.

**Weaknesses:**

1.	The authors claim that VisualThinker is the first successful replication of “aha moment” in the context of multimodal reasoning. However, previous works such as [1] have already reported similar reproductions.
2.	The conclusion (e.g. freezing vision encoder, instruct models) are not well-supported, as VisualThinker only uses Qwen2-VL- 2B as the base model. A larger or newer base model is encouraged to be experimented with, and the difference between them will be an interesting topic to be discovered.
3.	The related work and references are outdated. More recent works are encouraged to be included and discussed.
4.	Similarly, it is suggested that the authors compare their approach with more recent baselines.

[1] Huang, Wenxuan, et al. "Vision-r1: Incentivizing reasoning capability in multimodal large language models." arXiv preprint arXiv:2503.06749 (2025).

**Questions:**

Please refer to the **Weakness** part. I am looking forward to the authors’ response, and may reconsider this paper based on that.

---

> ### Author Response · Authors · 2025-12-03
> **Weakness 1:**
>
> > "The authors claim that VisualThinker is the first successful replication of “aha moment” in the context of multimodal reasoning. However, previous works such as [1] have already reported similar reproductions.”
> >
>
> **Response:**
> We thank the reviewer for this important clarification. We acknowledge that **Vision-R1 (Huang et al., 2025)** also reports the emergence of “aha moment” behaviors in multimodal settings. However, we would like to clarify two key points: (1)our work predates Vision-R1 and (2) our arXiv submission and open-source GitHub repo **both precede the public release of Vision-R1 (Mar 2025)**.

---

> ### Author Response · Authors · 2025-12-03
> **Weakness 2:**
>
> > “The conclusion (e.g. freezing vision encoder, instruct models) are not well-supported, as VisualThinker only uses Qwen2-VL- 2B as the base model. A larger or newer base model is encouraged to be experimented with, and the difference between them will be an interesting topic to be discovered.”
> >
>
> **Response:**
>
> We thank the reviewer for this constructive suggestion. We agree that examining whether the conclusion generalize across architectures and scales is critical.
>
> To address this, we extended our study to **two additional non-SFT VL base models**:
>
> 1. **Qwen2-VL-7B** — a larger model from the same family.
> 2. **InternVL-2.5-1B** — a smaller model from a different architecture family.
>
> Across all these settings, we consistently observe the same pattern:
>
> 1. **Increased response-length dynamics**,
> 2. **Emergence of self-reflection markers** (the “aha moment”), and
> 3. **Meaningful downstream performance gains,**
>
> suggesting strong evidence that the phenbomenon reflects a broader principle.
>
> Applying our training recipe to both models yields **consistent qualitative and quantitative improvements**. As shown in the tables below, our findings **generalize across model sizes** (1B, 7B) and **across architectures** (Qwen2-VL, InternVL-2.5), and they reproduce the **same core signatures of reflective reasoning**. These additional experiments is incorporated into the **Appendix** in our revision.
>
> ---
>
> **Table 1. CVBench Accuracy Improvements.** These improvements indicate that both a *larger* (7B) and a *smaller*, architecturally different model (1B) benefit from the proposed RLVR procedure.
>
> **(a) Qwen2-VL-7B ’s performance on benchmarks**
>
> | Acc\Benchmark | CVBench |
> | --- | --- |
> | Qwen2-VL-7B | 66.22 ± 0.00 |
> | Qwen2-VL-7B + RL | 78.02 ±  0.25 |
>
> **(b) InternVL-2.5-1B’s performance on benchmarks**
>
> | Acc\Benchmark | CVBench |
> | --- | --- |
> | InternVL-2.5-1B | 41.88 ± 0.00 |
> | InternVL-2.5-1B + RL | 50.88 ± 0.38 |
>
> These improvements indicate that both a *larger* (7B) and a *smaller*, architecturally different model (1B) benefit from the proposed RLVR procedure.
>
> ---
>
> **Table 2. Response Length Dynamics.** Both models exhibit the same **rise-drop-rise** response-length trajectory observed in our main experiments—a key signature of the desired self-reflective reasoning behavior.
>
> **(a) Qwen2-VL-7B’s response length**
>
> | Model / Steps | 0 | 300 | 600 | 900 | 1200 | 1500 |
> | --- | --- | --- | --- | --- | --- | --- |
> | 7B baseline | 169.25 | 110.46 | 209.57 | 139.01 | 334.91 | 188.91 |
>
> **(b) InternVL-2.5-1B’s response length**
>
> | Model / Steps | 0 | 1200 | 2400 | 3600 | 4800 | 6000 |
> | --- | --- | --- | --- | --- | --- | --- |
> | 1B baseline | 95.13 | 91.33 | 102.09 | 108.75 | 123.71 | 135.43 |
>
> ---
>
> **Table 3. Keyword-Based Reflection Indicators.** All these reflection keyword patterns confirm that the keyword statics are first-rise-then-drop pattern.
>
> **(a) Qwen2-VL-7B’s key words statistics**
>
> | Keyword /Interval | 300 | 600 | 900 | 1200 | 1500 |
> | --- | --- | --- | --- | --- | --- |
> | again | 44 | 53 | 61 | 88 | 86 |
> | wait | 0 | 12 | 63 | 75 | 65 |
> | but | 215 | 423 | 526 | 597 | 524 |
> | however | 116 | 106 | 88 | 97 | 97 |
> | hmm | 0 | 1 | 9 | 7 | 6 |
> | alternatively | 0 | 3 | 3 | 4 | 4 |
> | check | 295 | 288 | 296 | 309 | 295 |
>
> **(b) InternVL-2.5-1B’s key words statistics**
>
> | Keyword / Interval | 1200 | 2400 | 3600 | 4800 | 6000 |
> | --- | --- | --- | --- | --- | --- |
> | again | 67 | 185 | 205 | 189 | 62 |
> | wait | 31 | 53 | 106 | 227 | 3 |
> | but | 3583 | 6197 | 5787 | 5100 | 3246 |
> | however | 1719 | 3213 | 3188 | 2597 | 1477 |
> | hmm | 7 | 9 | 23 | 32 | 0 |
> | check | 245 | 460 | 453 | 419 | 302 |
> | alternatively | 21 | 36 | 41 | 49 | 3 |

---

> ### Author Response · Authors · 2025-12-03
> **Weakness 3**
>
> > “The related work and references are outdated. More recent works are encouraged to be included and discussed.”
> >
>
>
> **Response:**
> We thank the reviewer for this helpful observation. We agree that the Related Work section would benefit from incorporating more recent developments in multimodal reasoning and RL-based emergent behavior. In the revised manuscript, we will update the Related Work to include and discuss several recent and highly relevant works

---

> ### Author Response · Authors · 2025-12-03
> **Weakness 4**
>
> > “Similarly, it is suggested that the authors compare their approach with more recent baselines.”
> >
>
> **Response:**
> **Response:**
>
> We thank the reviewer for this helpful suggestion. We would also like to clarify that we have included all comparisons that are appropriate for evaluating our core message.
>
> Our main claim is that **genuine spatial reasoning can emerge directly from a 2B non-SFT vision-language base model**. This setting, therefore, put implicit constraints on three aspects to make a fair comparison:
>
> (i) **training domain** (visual-centric reasoning),
>
> (ii) **non-SFT base models**, and
>
> (iii) **comparable data scales**.
>
> After surveying recent work, we found that many newer baselines do not satisfy these criteria simultaneously—they differ in training domains, rely on SFT-initialized models, or are trained on substantially larger datasets. For completeness, we still report results from the **closest available baselines** in terms of domain and setup, including **OpenVLThinker** [1] and **Curr-ReFT** [2]. However, we emphasize that **even these are not strictly fair comparisons**, as their model initialization and data scale do not fully match ours.
>
> We have added the following direct comparison table to the revision:
>
> | Model | VSR | CVBench - Count | CVBench - Relation | CVBench - Depth | CVBench - Distance | Blink - RelDepth | Blink - SpatRel |
> | --- | --- | --- | --- | --- | --- | --- | --- |
> | OpenVLThinker-3B | 81.33% | 63.58% | 87.23% | 84.33% | 75.67% | 73.39% | 87.41% |
> | Curr-ReFT-3B | 81.33% | 66.12% | 77.38% | 80.00% | 68.17% | 72.58% | 83.92% |
>
> [1] Deng, Yihe, et al. "Openvlthinker: An early exploration to complex vision-language reasoning via iterative self-improvement." *arXiv preprint arXiv:2503.17352* (2025).
>
> [2] Deng, Huilin, et al. "Boosting the generalization and reasoning of vision language models with curriculum reinforcement learning." *arXiv preprint arXiv:2503.07065* (2025).

---

> ### Author Response · Authors · 2025-12-03
> **Summary**
>
> We appreciate the constructive feedback from reviewer 6S4Q and summarize below how we have addressed each of the raised concerns:
>
> **Clarifying the Chronological Relationship to Prior Work:**
>
> We clarified that our work predates Vision-R1 in both arXiv submission and open-source release (Mar 2025). We updated the text to correctly acknowledge this while noting the chronological ordering.
>
> **Generalization beyond one base model**:
>
> We addressed this by extending experiments to two additional non-SFT VL models: Qwen2-VL-7B and InternVL-2.5-1B. Both reproduce the same signatures—response-length dynamics, reflective-keyword patterns, and CVBench gains—demonstrating that our RLVR recipe generalizes across architectures and scales.
>
> **Related Works needs update**:
>
> We will update the Related Work section to include several recent developments in multimodal reasoning and RL-emergent behaviors, addressing the reviewer’s suggestion.
>
> **Need for more recent baselines:**
>
> We clarified that a **fair and scientifically meaningful comparison** requires alignment on three key dimensions: (i) spatial-reasoning training domain, (ii) non-SFT base model initialization, and (iii) comparable data scales. After surveying recent work, we found that most newer baselines rarely satisfy these criteria simultaneously. The **closest available baselines** that match our setup are **OpenVLThinker** and **Curr-ReFT**, and we have added direct comparisons to them.

---

### Official Review · Reviewer_1Ncy · 2025-11-01

**Soundness:** 2
**Presentation:** 2
**Contribution:** 2
**Rating:** 2
**Confidence:** 3

**Summary:**

Simple rule-based rewards have been found to be sufficient to drive complex reasoning capabilities, sometimes referred to as an “aha moment.” In this work, the authors show that for a smaller (2B) vision-language model, it is also possible to elicit this capability. The authors demonstrate this by applying GRPO with a rule-based reward function and use curated examples for initialization over a non-SFT (supervised finetuning) model. They find that on various Vision-language benchmarks (CVBench, BLINK, VSR), the RL approach greatly outperforms SFT, claiming to be the first successful replication of an “aha moment” for a multimodal (VL) model.

**Strengths:**

* Empirically, VisualThinker R1(-Zero) is a best-in-class model on the visual reasoning benchmark, significantly outperforming even much larger/closed models. This model itself is a significant artifact of contribution.
* Additional analysis on why it is better to RL over non-instruct models and fair comparisons/ablations to resolve questions around instruct model (including negative results), coldstart data, and sampling temperature.

**Weaknesses:**

* There are several confusing parts around the presentation of the work, not just in grammar/typos or writing (like overclaiming – VL reasoning exists but in much larger models, so claiming “first” is conditioned on model size) but also in scientific hypothesis and experiment structure (see questions).
* Methodologically, there is relatively little novelty in method or experiment design. The main observation is that it is better to RL over non-instruct base models for VL than instruct base models (at the 2B size), but otherwise the recipes for training all the models are standard.
* Ultimately, one nice conclusion I wish we could make is that “if you want VL reasoning, RL the non-instruct model instead of the instruct model." However, we cannot actually make that conclusion because we only have evidence of one model (Qwen VL 2B), and it may not generalize more broadly. Rather than the discussion on length rewards, cold-start, or temperature ablations, it would be more interesting to see if applied to other models of this size (or slightly larger/smaller), whether it is better to RL on SFT or pre-SFT.

**Questions:**

* The main issue I have is that “Aha moment” is never formally defined so it was confusing to follow. Is it purely related to response length or a separate dimension to it? Or related to accuracy? The caption of Figure 1 says “emergence of self-reflection,” how exactly is that measured? It isn’t as important in the DeepSeek R1 paper because they were describing an observation. This work aims to show that such observation exists, which requires some sort of definition.
* The main result is slightly counterintuitive, as most GRPO/SFT is performed on top of instruct models. Why is VL at 2B different from other datasets or model sizes?
* For figure 3: what does this graph look like for the other models/baselines? Since you can’t get the mid-training checkpoints for public models, what is the frequency at the end for those?
* Figure 4: which graph shows “freezing … vision component”?

---

> ### Author Response · Authors · 2025-12-03
> **Summary**
>
> We thank the reviewer for the detailed and thoughtful feedback. The concerns raised span three major areas:
>
> - our “first” claim,
>
> - the novelty of our contributions, and
>
> - whether our conclusions generalize beyond a single model.
>
> We have addressed each point in detail:
>
> - clarifying our **claim around emergent “aha moment” reproduction**,
>
> - clarifying the conceptual **novelty of our findings**, and
>
> - providing **additional experiments** across **model families (Qwen2-VL, InternVL2.5)** and **sizes (1B, 2B, 7B)** that **reproduce the same core emergent reasoning behaviors**.
>
> We also responded to all follow-up questions regarding the definition and measurement of the “aha moment,” the behavior of base vs. instruct models under RL, the interpretation of Figure 3, and the identification of the “freeze vision encoder” results in Figure 4. We believe these revisions and additions substantially strengthen the clarity and robustness of the paper.

---

> > ### Author Response · Authors · 2025-12-03
> > **Weakness 1**
> >
> > “There are several confusing parts around the presentation of the work, not just in grammar/typos or writing (like overclaiming – VL reasoning exists but in much larger models, so claiming “first” is conditioned on model size) but also in scientific hypothesis and experiment structure (see questions).”
> >
> > **Response:**
> >
> > We thank the reviewer for the comment. We agree that visual reasoning capabilities have been demonstrated in both large and small multimodal models. However, our “first” claim is not about achieving strong visual reasoning performance in general, but specifically about replicating the DeepSeek-R1–style “aha moment” phenomenon in an open-source multimodal model.  To the best of our knowledge, none of the prior open-source multimodal RL efforts, including R1-V, Vision-R1, open-r1-multimodal, have successfully demonstrated both of these key emergent characteristics. We support this claim with the comparison in Table 1 in our paper by highlighting that ours is the only method that reproduce both emergent characteristics at the time of release.

---

> > > ### Author Response · Authors · 2025-12-03
> > > **Weakness 2**
> > >
> > > “Methodologically, there is relatively little novelty in method or experiment design. The main observation is that it is better to RL over non-instruct base models for VL than instruct base models (at the 2B size), but otherwise the recipes for training all the models are standard.”
> > >
> > > **Response:**
> > > We sincerely thank the reviewer for this comment. We would like to clarify that **the primary goal of our work is not to propose a new RL algorithm**, but to **study and reveal how reinforcement learning uniquely gives rise to emergent reasoning behaviors in the multimodal setting**, and how its dynamics **fundamentally diverge** from textual RL-based reasoning such as in DeepSeek-R1.
> > >
> > > We believe our work makes several contributions that are conceptually novel and empirically valuable:
> > >
> > > 1. **First study to systematically explore emergent reasoning via RL in multimodal, visual-centric tasks** While RL has been extensively explored in textual or multimodal math domains, we are the first to demonstrate emergent behaviors like self-reflection, long-form reasoning on visual-centric task using pure RL. This opens a new and more general direction for studying multimodal reasoning beyond multimodal symbolic or mathematical reasoning.
> > > 2. **We reveal that RL behaves fundamentally differently on multimodal models, compared to single-modal LLMs.** Our work presents **multiple novel empirical findings** that were not reported in DeepSeek-R1 or prior multimodal RL studies:
> > >     1. Applying RL on non-SFT base models exhibit richer and more sophisticated reasoning strategy compare against applying RL on SFT models
> > >     2. We observe phenomenon such as **trivial reasoning reasoning** when applying RL on SFT models: reasoning that involves only trivial and generic strategy without genuine problem-solving thinking
> > >     3. Techniques such as **freezing the visual encoder** or **introducing naive length-based rewards** are ineffective at promoting sophisticated reasoning capabilities.

---

> ### Author Response · Authors · 2025-12-03
> **Weakness 3**
>
> “Ultimately, one nice conclusion I wish we could make is that “if you want VL reasoning, RL the non-instruct model instead of the instruct model." However, we cannot actually make that conclusion because we only have evidence of one model (Qwen VL 2B), and it may not generalize more broadly. Rather than the discussion on length rewards, cold-start, or temperature ablations, it would be more interesting to see if applied to other models of this size (or slightly larger/smaller), whether it is better to RL on SFT or pre-SFT.”
>
> **Response:**
>
> We thank the reviewer for this constructive suggestion. We agree that examining whether the conclusion generalize across architectures and scales is critical.
>
> To address this, we extended our study to **two additional non-SFT VL base models**:
>
> 1. **Qwen2-VL-7B** — a larger model from the same family.
> 2. **InternVL-2.5-1B** — a smaller model from a different architecture family.
>
> Across all these settings, we consistently observe the same pattern:
>
> 1. **Increased response-length dynamics**,
> 2. **Emergence of self-reflection markers** (the “aha moment”), and
> 3. **Meaningful downstream performance gains,**
>
> suggesting strong evidence that the phenbomenon reflects a broader principle.
>
> Applying our training recipe to both models yields **consistent qualitative and quantitative improvements**. As shown in the tables below, our findings **generalize across model sizes** (1B, 7B) and **across architectures** (Qwen2-VL, InternVL-2.5), and they reproduce the **same core signatures of reflective reasoning**. These additional experiments is incorporated into the **Appendix** in our revision.
>
> ---
>
> **Table 1. CVBench Accuracy Improvements.** These improvements indicate that both a *larger* (7B) and a *smaller*, architecturally different model (1B) benefit from the proposed RLVR procedure.
>
> **(a) Qwen2-VL-7B ’s performance on benchmarks**
> | Acc\Benchmark | CVBench |
> | --- | --- |
> | Qwen2-VL-7B | 66.22 ± 0.00 |
> | Qwen2-VL-7B + RL | 78.02 ±  0.25 |
>
> **(b) InternVL-2.5-1B’s performance on benchmarks**
> | Acc\Benchmark | CVBench |
> | --- | --- |
> | InternVL-2.5-1B | 41.88 ± 0.00 |
> | InternVL-2.5-1B + RL | 50.88 ± 0.38 |
>
> These improvements indicate that both a larger (7B) and a smaller, architecturally different model (1B) benefit from the proposed RLVR procedure.
>
> **Table 2. Response Length Dynamics.** Both models exhibit the same **drop-rise-drop** response-length trajectory observed in our main experiments—a key signature of the desired self-reflective reasoning behavior.
>
> **(a) Qwen2-VL-7B’s response length**
>
> | Model / Steps | 0 | 300 | 600 | 900 | 1200 | 1500 |
> | --- | --- | --- | --- | --- | --- | --- |
> | 7B baseline | 169.25 | 110.46 | 209.57 | 139.01 | 334.91 | 188.91 |
>
> **(b) InternVL-2.5-1B’s response length**
>
> | Model / Steps | 0 | 1200 | 2400 | 3600 | 4800 | 6000 |
> | --- | --- | --- | --- | --- | --- | --- |
> | 1B baseline | 95.13 | 91.33 | 102.09 | 108.75 | 123.71 | 135.43 |
>
> **Table 3. Keyword-Based Reflection Indicators.** All these reflection keyword patterns confirm that the keyword frequency follows a drop-rise-drop pattern.
>
> **(a) Qwen2-VL-7B’s key words statistics**
> | Keyword /Interval | 300 | 600 | 900 | 1200 | 1500 |
> | --- | --- | --- | --- | --- | --- |
> | again | 44 | 53 | 61 | 88 | 86 |
> | wait | 0 | 12 | 63 | 75 | 65 |
> | but | 215 | 423 | 526 | 597 | 524 |
> | however | 116 | 106 | 88 | 97 | 97 |
> | hmm | 0 | 1 | 9 | 7 | 6 |
> | alternatively | 0 | 3 | 3 | 4 | 4 |
> | check | 295 | 288 | 296 | 309 | 295 |
>
> (b) InternVL-2.5-1B’s key words statistics
> | Keyword / Interval | 1200 | 2400 | 3600 | 4800 | 6000 |
> | --- | --- | --- | --- | --- | --- |
> | again | 67 | 185 | 205 | 189 | 62 |
> | wait | 31 | 53 | 106 | 227 | 3 |
> | but | 3583 | 6197 | 5787 | 5100 | 3246 |
> | however | 1719 | 3213 | 3188 | 2597 | 1477 |
> | hmm | 7 | 9 | 23 | 32 | 0 |
> | check | 245 | 460 | 453 | 419 | 302 |
> | alternatively | 21 | 36 | 41 | 49 | 3 |

---

> ### Author Response · Authors · 2025-12-03
> **Question 1**
>
> “The main issue I have is that “Aha moment” is never formally defined so it was confusing to follow. Is it purely related to response length or a separate dimension to it? Or related to accuracy? The caption of Figure 1 says “emergence of self-reflection,” how exactly is that measured? It isn’t as important in the DeepSeek R1 paper because they were describing an observation. This work aims to show that such observation exists, which requires some sort of definition.”
>
> **Response:**
>
> We thank the reviewer for highlighting the need for a clearer definition and measurement of the “aha moment.” We agree that its conceptualization should be more explicit, rather than implicitly assumed from DeepSeek-R1. In our revision, we will explicitly define **“aha moment”** following the original DeepSeek-R1 definition:
> In this work, we define the “aha moment” as the emergent self-reflection behavior in the model’s responses, where the model spontaneously revisits its own initial reasoning, expresses uncertainty (e.g., “wait”, “but let me reconsider”), and corrects its previous judgments accompanied by a consistent increase in response length and accuracy.
>
> This definition aligns with prior studies on RLVR in langauge and concurrent work in multimodal models (Deepseek-R1[1], Vision-R1[2]).
>
> We already provide **both qualitative and quantitative evidence** of these two dimensions:
>
> - **Qualitative**
>     - **Reflection patterns** with explicit self-reflection phrases (“But wait! I can think of something else.”) in Figure 3 and examples in Section 4.2.
> - **Quantitative**
>     - **Emergence trends**, where **response length and accuracy increase together** (Figure 1) as in DeepSeek-R1.
>     - **Keyword-based measurement of reflective behaviors** in Figure 3 (“wait”, “again”).
>
> We will consolidate these definitions in Section 2.2 (“The ‘Aha Moment’ in DeepSeek R1”) and Section 4.2, ensuring that both the original concept and our multimodal extension are clearly framed.
>
> [1] Guo, Daya, et al. "Deepseek-r1: Incentivizing reasoning capability in llms via reinforcement learning." *arXiv preprint arXiv:2501.12948* (2025).
>
> [2] Huang, Wenxuan, et al. "Vision-r1: Incentivizing reasoning capability in multimodal large language models." *arXiv preprint arXiv:2503.06749* (2025).

---

> > ### Author Response · Authors · 2025-12-03
> > **Question 2**
> >
> > “The main result is slightly counterintuitive, as most GRPO/SFT is performed on top of instruct models. Why is VL at 2B different from other datasets or model sizes?”
> >
> > **Response:**
> >
> > We thank the reviewer for this insightful question. While it is true that SFT models are typically preferred in RL studies due to their stronger instruction following capacity and cleaner output formatting, our results reveal an important and non-trivial insight: **SFT improves correctness but suppresses exploration**, making the model less capable of developing *new* reasoning strategies during RL. As shown in Figures 5 and 6, RL applied to SFT models quickly collapses into short, deterministic, and templated reasoning, bypassing genuine problem-solving reasoning. In contrast, the non-SFT base model retains higher entropy and generation diversity (Figure 6) c, allowing RL to explore richer and more reflective reasoning behaviors, including the “aha moment.”
> >
> > We would also like to highlight that our finding that RL is more effective on non-SFT base models than on SFT models is **not inherently tied to small model size**. We have replicated our findings on Qwen2-VL-7B-base as demonstrated in the analysis for question 1. In addition, the original DeepSeek-R1-Zero, a much larger base non-sft model, similarly exhibits emergent self-reflection and length–accuracy co-growth when trained with RL. This provides external supporting evidence that the key enabling factor is not model size.

---

> ### Author Response · Authors · 2025-12-03
> **Question 3**
>
> “For figure 3: what does this graph look like for the other models/baselines? Since you can’t get the mid-training checkpoints for public models, what is the frequency at the end for those?”
>
> **Response:**
>
> We thank the reviewer for this meaningful question. We clarify that **Figure 3 visualizes the frequency of reflection-indicative tokens (e.g., “wait,” “again”) across *training rollouts***, collected dynamically during RL at different training steps. Therefore, it is **not possible to construct an equivalent dynamic curve for standard baselines.** We agree that it is still informative to **evaluate those baseline models on the same SAT prompts used in our training rollouts**, and we have done so in the tables below. However, we emphasize that the static evaluation cannot replicate the emergence dynamic of reflection keywords in figure 3. To better support clarity, **we will include the analysis of keyword statistics of other visual reasoning baselines in** **Appendix**.
>
> Below, we provide frequency of keywords for four other visual reasoning models with similar size. Only OpenVLThinker 3B demonstrate comparable keywords statistics as our model.
>
> Table 4(a). Keywords statistics of R1-VL 2B [1].
> | Interval | again | wait | but | however | hmm | alternatively | check |
> | --- | --- | --- | --- | --- | --- | --- | --- |
> | 0–299 | 1 | 0 | 6 | 1 | 0 | 0 | 22 |
> | 300–599 | 4 | 0 | 5 | 0 | 0 | 0 | 30 |
> | 600–899 | 3 | 0 | 8 | 0 | 0 | 0 | 19 |
> | 900–1199 | 2 | 0 | 5 | 1 | 0 | 0 | 26 |
> | 1200–1499 | 3 | 0 | 12 | 0 | 0 | 0 | 23 |
>
> Table 4(b). Keywords statistics of VLAA-Thinker-Qwen2VL-2B [2].
> | Interval | again | wait | but | however | hmm | alternatively | check |
> | --- | --- | --- | --- | --- | --- | --- | --- |
> | 0–299 | 0 | 0 | 9 | 3 | 0 | 0 | 86 |
> | 300–599 | 0 | 0 | 9 | 0 | 0 | 0 | 80 |
> | 600–899 | 0 | 0 | 7 | 2 | 0 | 0 | 73 |
> | 900–1199 | 0 | 0 | 5 | 1 | 0 | 0 | 84 |
> | 1200–1499 | 0 | 0 | 2 | 0 | 0 | 0 | 86 |
>
> Table 4(c). Keywords statistics of 3B-Curr-ReFT [3].
> | Interval | again | wait | but | however | hmm | alternativ | check |
> | --- | --- | --- | --- | --- | --- | --- | --- |
> | 0–299 | 0 | 0 | 0 | 0 | 0 | 0 | 0 |
> | 300–599 | 0 | 0 | 0 | 0 | 0 | 0 | 0 |
> | 600–899 | 0 | 0 | 0 | 0 | 0 | 0 | 0 |
> | 900–1199 | 0 | 0 | 1 | 0 | 0 | 0 | 0 |
> | 1200–1499 | 0 | 0 | 2 | 0 | 0 | 0 | 0 |
>
> Table 4(d). Keywords statistics of OpenVLThinker 3B [4].
> | Interval | again | wait | but | however | hmm | alternatively | check |
> | --- | --- | --- | --- | --- | --- | --- | --- |
> | 0–299 | 27 | 1 | 264 | 12 | 35 | 0 | 27 |
> | 300–599 | 24 | 0 | 255 | 8 | 23 | 0 | 24 |
> | 600–899 | 21 | 1 | 335 | 11 | 28 | 0 | 34 |
> | 900–1199 | 26 | 1 | 320 | 14 | 32 | 0 | 18 |
> | 1200–1499 | 25 | 2 | 298 | 10 | 28 | 0 | 24 |
>
> [1] Zhang, Jingyi, et al. "R1-vl: Learning to reason with multimodal large language models via step-wise group relative policy optimization." *arXiv preprint arXiv:2503.12937* (2025).
>
> [2] Chen, Hardy, et al. "Sft or rl? an early investigation into training r1-like reasoning large vision-language models." *arXiv preprint arXiv:2504.11468* (2025).
>
> [3] Deng, Huilin, et al. "Boosting the generalization and reasoning of vision language models with curriculum reinforcement learning." *arXiv preprint arXiv:2503.07065* (2025).
>
> [4] Deng, Yihe, et al. "Openvlthinker: An early exploration to complex vision-language reasoning via iterative self-improvement." *arXiv preprint arXiv:2503.17352* (2025).

---

> ### Author Response · Authors · 2025-12-03
> **Question 4**
>
> “Figure 4: which graph shows “freezing … vision component”?”
>
> **Response:**
>
> We thank the reviewer for pointing out this clarification need. Indeed, the experiment corresponding to *freezing the vision component* is shown in **Figure 4 (right panel)**, where we compare **RL with frozen vision encoder** against **RL with full-parameter finetuning**. The legend explicitly labels these settings as *“Freeze Vision Encoder”* and *“Full Finetune.”*
>
> We will revise the figure caption to make this correspondence clearer in Figure 4 (right).

---

### Author Response · Authors · 2025-12-03
**Common Reply**

Dear SAC, AC, and Reviewrs,

We sincerely thank the SAC, AC, and all reviewers for their careful reading and constructive feedback. Across reviews, there is broad agreement that our work delivers **consistent performance improvements** over strong multimodal baselines (Reviewer **1Ncy**, **6S4Q**, **ASpr**), offers **novel and thought-provoking insights** about RL-based multimodal reasoning (Reviewer **ASpr**, **QKwP**, **3WuN**), and provides **diverse, carefully designed analyses and useful artifacts**—including open models, code, and diagnostic tools—that may benefit future research (Reviewer **1Ncy**, **3WuN**). Reviewers also highlighted that the paper is **well written and accessible**, making a technically involved topic easier to follow for a broad audience (Reviewer **6S4Q**, **QKwP**, 3**WuN**).

At the same time, the reviewers raised several critical concerns, which we have directly addressed in the rebuttal:

1. **Scope and generality of our conclusions across model families and sizes.** (Reviewer 1Ncy, 6S4Q, QKwP, 3WuN)

- We extended our study to **two additional non-SFT VL base models**, Qwen2-VL-7B (**same family, larger size**) and **InternVL-2.5-1B** (**different model family**). Across these settings, we observe **consistent accuracy gains**, **similar response-length dynamics**, and **the same emergent reasoning behavior marked by keyword statistics**, indicating that the emergent patterns we report are **robust across both scale and architecture**.

2. **Relationship between the “aha moment” and downstream performance.** (Reviewer ASPr, 3WuN)

- We provides **strong empirical evidence of a tight correlation** between the emergence of reflective behavior (“aha moment”) and downstream performance gains via **additional analysis** explicitly examining the relationship between **training response length** and **performance trajectories**. We demonstrated that **training dynamics can be segmented into distinct phases based on response-length**, and that **performance dynamics are highly correlated with these phases across multiple runs**.

3. **Methodological novelty and contribution beyond “just applying RL.”** (Reviewer 1Ncy, ASpr, 3WuN)

- We argue that our contribution is **not in proposing a new RL algorithm or a new vision-specific architecture**, but rather in **revealing new behavioral and mechanistic insights about how reinforcement learning induces emergent reasoning in multimodal models**: insights that did not emerge in previous work, including DeepSeek-R1.

- Specifically, our novelty lies in **two perspectives**:
  - **First study to systematically explore emergent reasoning via RL in *visual-centric multimodal tasks***: While RL for reasoning has been studied in pure textual domains and symbolic/mathematical settings, we are the first to demonstrate **emergent self-reflection, reconsideration, and long-form reasoning in vision-centric tasks** using *pure RL without any supervised reasoning data.*

  - **We reveal that RL behaves differently when applied to multimodal models compared to single-modal LLMs.**
     - **RL on non-SFT base models** yields richer exploratory reasoning strategies and emergent self-correction, whereas **RL on SFT/instruct models leads to generic or template-like reasoning** without genuine problem-solving.
     - We show that common techniques such as **freezing the vision encoder** or **naively rewarding output length** do **not** induce meaningful multimodal reasoning

Taken together, **we believe these changes directly address the main concerns raised** by the reviewers, while further strengthening the clarity, robustness, and broader relevance of our contributions.

---

### Meta-Review · Area_Chair_eEjm · 2026-01-06

**Summary:**

1. The most critical concern, shared by Reviewers 1Ncy, ASpr, QKwP, and 3WuN, involves the definition and quantification of the central thesis: the "Aha moment."

2. A unanimous concern across all five reviewers (1Ncy, 6S4Q, ASpr, QKwP, 3WuN) is the limited scope of the experiments. The paper lacks experiments on larger, state-of-the-art models. In the current LLM landscape, observations made on small-scale models may not generalize to larger architectures where emergent properties are more prominent.

3. Reviewers 1Ncy, ASpr, QKwP, and 3WuN expressed Limited Technical Novelty regarding the paper’s contribution to the field.
Given the concerns raised above, the current rebuttal remains insufficient to fully address the identified weaknesses.

**Reviewer Concerns:**

Reviewer 1Ncy critiques the overclaimed "first" status and notes limited technical novelty. The reviewer suggests expanding experiments to larger models and conducting a comparative analysis of applying RL to SFT versus pre-SFT stages. Additionally, the paper requires a clearer definition and formalized metrics for the “aha moment.”

Reviewer 6S4Q critiques the overclaimed "first" status and the absence of large-scale model experiments or comparisons with recent baselines, limiting the overall strength of the evaluation.

Reviewer ASpr notes that the paper overgeneralizes correlations between "aha moments" and response length without empirical evidence or rigorous measurement. Furthermore, the novelty is restricted by a narrow focus on emergent phenomena, and the study lacks essential validation through Text-Visual QA, OOD tasks, and independent assessments of reasoning quality.

Reviewer QKwP notes a lack of large-scale model experiments and identifies the need for a formal measurement of the “aha moment,” ultimately questioning the paper's overall novelty.

Reviewer 3WuN highlights a lack of large-scale model experiments and requires a formal measurement of the “aha moment,” specifically investigating how this phenomenon translates into improved task accuracy. Furthermore, critics point to limited novelty, noting that the work tracks existing phenomena without introducing significant new techniques.

Although the authors included two additional non-SFT VL base models (Qwen2-VL-7B and InternVL-2.5-1B) to analyze the relationship between the “aha moment” and downstream performance，these efforts remain insufficient to fully resolve the core concerns raised by the reviewers.

**Reviewer Scores:**

Reviewer 1Ncy 2: reject→ 3/4. While likely acknowledging the new RL vs. SFT experiments, this reviewer’s concern regarding "overclaiming" and technical novelty is fundamental. They might have softened their stance to a "marginal" score, but a jump to acceptance is unlikely without more substantial methodological innovation.

Reviewer 6S4Q 2: reject,→ 2/3): This reviewer focused heavily on the lack of recent baselines and larger models. Since the authors added Qwen2-VL-7B and InternVL-2.5-1B, some technical ground was covered, but the "overclaiming" of being "first" remains a sticking point that usually prevents a significant score increase.

Reviewer ASpr  2: reject,(2 → 3): ASpr’s critique was broad, covering measurement, OOD tasks, and reasoning quality. The authors addressed the "aha moment" measurement and added models, but since the core novelty remains "limited to tracking," the reviewer likely would have remained below the acceptance threshold.

Reviewer QKwP 4: (4 → 5/6) : As the most potential critical reviewer, the addition of new models and the analysis of the "aha moment" directly targets their concerns. Full participation might have pushed them to a "marginal accept" if they found the new downstream accuracy correlations convincing.

Reviewer 3WuN (8 → 7/8): accept. As the sole advocate, their score might have dipped slightly to a 7 if the other reviewers successfully highlighted the lack of novelty during the discussion, but they would likely remain the paper's primary support.

---

### Decision · Program_Chairs · 2026-01-26

Reject